# Epidemiology of bovine schistosomiasis and associated risk factors in Ethiopia: A systematic review with meta-analysis of published articles, 2008–2018

Melkie Dagnaw[1]*, Bihonegn Wodajnew[2], Tsegaw Fentie[3], Atsede Solomon[4], Firdyawukal Abuhay[5], Nigist Bizu[1], Ahmed Abi[6]

1 Department of Clinical Veterinary Medicine, College of Veterinary Medicine and Animal Sceince, University of Gondar, Gondar, Ethiopia, 2 Department of Veterinary Parasitology, College of Agriculture and Natural Sciences, Assosa University, Assosa, Ethiopia, 3 Department of Veterinary Epidemiology, College of Veterinary Medicine and Animal Sceince, University of Gondar, Gondar, Ethiopia, 4 Department of Veterinary Pharmacy, College of Veterinary Medicine and Animal Sceince, University of Gondar, Gondar, Ethiopia, 5 Department of Veterinary Microbiology, College of Veterinary Medicine and Animal Sceince, University of Gondar, Gondar, Ethiopia, 6 Faculty of Agriculture and Environmental, Department of Animal Production and Marketing, Sciences, Gulu University, Gulu, Uganda

* melkiedagnaw3528@gmail.com

**Data Availability Statement:** All relevant data are within the paper and its Supporting information files.

## Abstract

Schistosomiasis is a tropical and subtropical parasitic infection that affects both animals and humans. It's caused by the Schistosoma genus and spreads via snails as an intermediate host. Schistosoma bovis is widely spread in Ethiopia's Northern, Eastern, Southwestern, and Central regions. It is an economically significant cattle disease with global health implications. Despite numerous prevalence studies of bovine schistosomiasis in different regions of the country, no systematic review and meta-analysis of the disease has ever been undertaken. As a reason, the purpose of this research was to provide information that can be used in the planning and design of *Schistosoma* interventions in Ethiopia, as the world aims to eliminate schistosomiasis as a public health problem by 2030. The PRISMA statement guidline was used to check the eligiblities of the included studies. Electronic bibliographic databases such as PubMed, Google Scholar, HINARI, and Web of science, reference lists from retriv articles, books, libraries, megazins and existing reviews manaually were employed for literature searches. The pooled prevalence of bovine schistosomiasis and heterogeneity among included studies was estimated by employing random effect models and the inverse variance index, respectively. To investigate the source of heterogeneity across and within studies, subgroup analysis was undertaken based on sample size, study years, and study regions. To assess publication bias and small study effects, funnel plotsand Egger's regression test were used. The pooled prevalence was calculated with a 95% confidence interval using STATA 17 software. To identify the various risk variables related to the prevalence of bovine schistosomiasis, a pooled odd ratio was used Based on the inclusion criteria, a total of 20 studies were discovered and included in the meta-analysis. The prevalence of bovine schistosomiasis was found to range from 22 to 45.7%. In this meta-analysis, the estimated pooled prevalence of bovine schistosomiasis was 24% (95% CI: 17% to

**Funding:** The authors received no specific funding for this work.

**Competing interests:** The authors have declared that no competing interests exist.

31%). Substantial heterogeneity was observed across the included studies ($I^2$ = 88.04%; p< 0.001). The results of the funnel plot and Egger'stests revealed no substantial publication bias (Egger's test; p = 0.509). The pooled odds ratio indicated that poor body condition was positively associated with the infection rate of bovine schistosomiasis (OR: 4.915, 95% CI: 2.675 to 9.030; p<0.001), with poor body conditioned animals having 4.915 times higher likelihood to schistosomiasis infection. This review found that the overall prevalence of bovine schistosomiasis was high and the body condition of animals was identified risk factor for *Schistosoma* infection among the studied factors. Therefore, this review found that the occurrence of bovine schistosomiasis was strongly associated with the state of body condition of animals. Further, sufficient numbers of prospective studies should be conducted to address other potential risk factors of bovine schistosomiasis in Ethiopia.

## Introduction

Schistosomiasis is one of the fifteen neglected tropical diseases (NTD) because it is unusual among the helminth diseases for two reasons: much of the pathogenesis is due to the eggs (rather than larvae or adults), and most of the pathology is caused by the host immune system (Delayed-type hypersensitivity and granulomatous reactions) [1] and is the second parasitic disease in humans after malaria. Schistosomiasis is referred to as bilharziasis after a German physician named Theodor Bilharz, who originally documented the cause of urinary schistosomiasis in 1851 [2]. Schistosomiasis affects the productivity, reproductive performance, and medically important parasitic disease of humans and animals living in the circulatory system of definitive hosts [3].

The major *Schistosoma* species that have a significant impact on domestic and wild animals are found throughout the world's tropical and subtropical areas. These include the following: *Schistosoma bovis* parasitizes cattle, goats, and sheep in Africa and Southern Europe; *Schistosoma mattheeii* parasitizes sheep, cattle, and other domestic animals in southern Africa; *Schistosoma japonicum* parasitizes humans, cats, and mammals in Asia; *Schistosoma magrebowiei* parasitizes antelopes in Central Africa; and *Schistosoma rodhaini* parasitizes wild rodents and carnivores *Schistosoma spindale*, *Schistosoma indicum*, *Schistosoma incognitum*, *Schistosoma nasale*, and *Schistosoma leiperi* are other Schistosoma species of veterinary interest in Asia [4]. *Schistosoma bovis'* primary hosts are ovines, bovines, and caprines, with small wild ruminants serving as secondary hosts. *Schistosoma bovis* is found all over Africa, Southwest Asia, and Europe [5].

Bovine schistosomiasis is a slug-borne trematode infection of cattle caused by the genus *Schistosom*a. Schistosoma bovis is endemic to northern, southwestern, eastern, and central parts of Ethiopia. Schistosoma bovis develops a syndrome characterized by liver damage, rough coat, pale mucosa, severe emaciation and reduced reproductive capacity, leading to a major economic downturn and public health problem [6].

Parasitological, immunological and molecular approaches are used to diagnose Schistosoma. Immunodiagnostic techniques detect species-specific antigens or antibodies; Identification of parasite DNA with molecular methods using the nucleic acid sequence of schistosomal GSTs [7] and nucleic acid hybridization techniques using complementary DNA probes were retrieved.

The sedimentation technique is a gold standard approach to detecting fecal egg shedding in the field. Moisture, rainfall, temperature, bodies of water (stagnant ponds, swamps, streams,

rivers, irrigation canals, marshes and dams) and snails as intermediate hosts influence animal and human schistosomiasis [8]. In addition, animal immunity/status, sex, breed, season, age, and housing system all play a role in schistosomiasis infection rates in animals [9].

Various epidemiological studies on schistosomiasis in cattle have been carried out in Ethiopia. The prevalence of Schistosoma bovis has been studied by several authors at different times and in different parts of Ethiopia, and a prevalence of 12.3–29% has been reported by Kerie and Seyoum; Alemayehu and Asrat; Habtamu and Mariam; Aylate et al. [10–13] in different districts of the Amhara region. Since the occurrence of biotypes suitable for the development and reproduction of intermediate hosts is closely linked to the occurrence of stagnant water and swampy grazing land, these epidemiological studies are an indication of the regular occurrence of diseases, especially in the area of stagnant water and swampy grazing land.

The results of this systematic review can help professionals working in animal, human and environmental health fields to keep track of the problems of parasitic zoonoses at different levels. It also serves as collective basic information in implementing prevention and control measures of foodborne zoonoses related to parasitic infections and in designing research and/or surveillance. As a result, the aim of this systematic review and meta-analysis was to provide some information that can be used in planning and designing *Schistosoma* interventions in Ethiopia as the world aims to eliminate schistosomiasis as a public health problem by 2030.

## Methods

The PRISMA (Preferred Reporting Items for Systematic Reviews and Meta-analysis) checklist [14] (S1 Table) was used to conduct this review. The checklist is used to confirm the inclusion of relevant information for this review based on the underline protocols.

### Search strategy

The literature search was conducted between September 01, 2021, and April 28, 2022. To explore all relevant studies, a thorough search technique was devised. Two authors (TF and NB) independently reviewed abstracts and manuscripts for eligibility. Databases such as PubMed, Google Scholar, HINARI, Web of science, reference listes from retriv articles and existing reviews manaually(open google internet, even libraries magazine, chapters, and books) searched for adddditional studies were used for literature searches to select included studies. This systematic review and meta-analysis used the CoCoPop (Condition, Context, and Population) framework to search for relevant articles. The condition was schistosomiasis (Co), the context was Ethiopia (Co), and the Population was cattle (Pop). The PubMed search strategy included Medical Subject Heading (MeSH) terms and a range of important keywords.

Epidemiologically schistosomiasis is a worldwide infection, which infects animals and humans. The research question was "what is the prevalence of schistosomiasis and its associated risk factors among cattle in Ethiopia?" During an online search, the Boolean operators "AND / OR" were employed to find relevant findings by combining similar phrases /words. Searching terms used were (schistosomiasis OR schistosomiasis infection OR bovine schistosomiasis) AND (epidemiology OR prevalence OR infection rate) AND (cattle OR bovine OR animals infected) and (risk factors OR predisposing factors) AND (Ethiopia). A restriction was placed on the language of publication is English. All identified studies were imported to End-Note 20 software to remove duplicates and scientific citations of the references.

### Study eligibility

The studies were organized in different sections; firstly arranged systematically based on the title, main objective, and content of the abstract. The second stage involved checking the type

of article against a list of inclusion and exclusion criteria. The crucial points to include relevant studies were; (i) all studies have been cross-sectional study design and employed both random and non-random sampling technique, (ii) clear estimation of prevalence, and (iii) the selection included only prevalence of schistosomiasis on bovine species, (iv) Study animals and the setting described in detail (v) year of study from 2008–2018 conducted in Ethiopia were considered. Articles were independently selected by two reviewers (MD and AS) and agreed on the selection of eligible studies and reached consensus on which studies met the inclusion criteria. Outbreak reports, case series, traditional reviews, cohort studies papers and case-control studies, and experimental (clinical trial) studies were excluded. In general, these inclusion and exclusion criteria were used for data extraction and a meta-analysis (study screening strategy and reasons for exclusion) as presented in Fig 1.

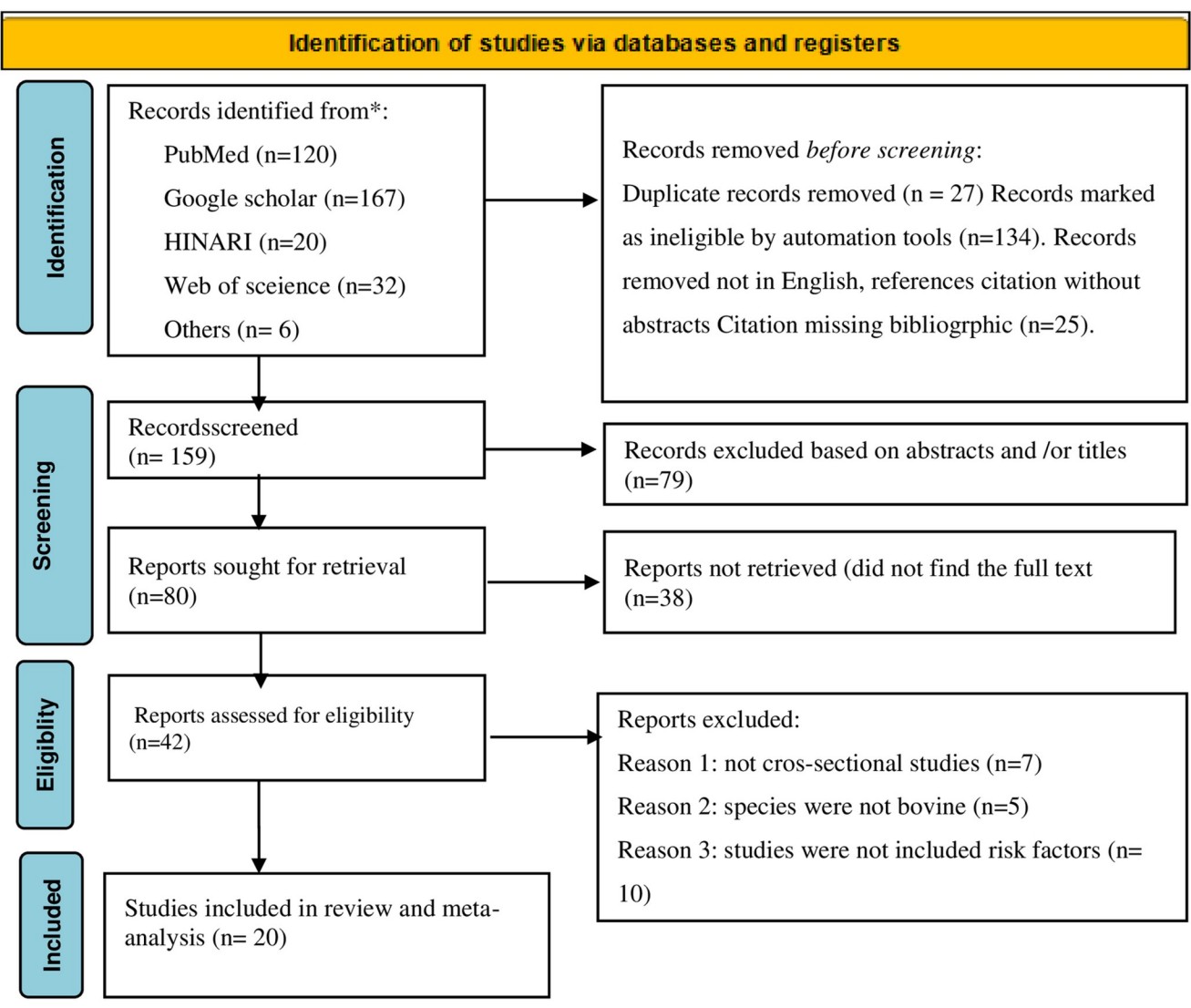

**Fig 1. PRISMA flow diagram for selecting included studies in Ethiopia from 2008 to 2018 years.**

## Data extraction

Data extraction was performed independently by two authors (NB and BW). Any discrepancies were resolved by consensus in consultation with the lead authors (TF and AA). The following information was gathered from the included studies, as shown in (Table 1). The name of the first author and publication year, year of study, study design, study regions and geographical location, sampling design, sample size, event and non-event, and prevalence of schistosomiasis among cattle population.

## Study quality assessment

Quality assessment was done in this review to check the methodological quality of this systemeti review. Meta-analysis results are dependent on the quality of the included studies. The literature search was performed by two researchers (FA and MD) independently using the quality assessment tool (AMSTAR-2) [15] (S1 File). The Critical Appraisal Quality Assessment Tool consists of 15 items (questions) involving randomized or non-randomized trials of health interventions, or both. Some of the quality assessment questions are used for both randomized and non-randomized trials. Others are only intended for randomized or non-randomized trials. For example the first question. Do the research questions and inclusion criteria for the review include the components of PICO? It is only used for randomized trials of health interventions. This systematic review and meta-analysis included non-randomized studies (observational study). The research question includes components of condition, context and population (CoCoPop).

**Table 1. Descriptive summary included studies of the prevalence of bovine schistosomiasis in Ethiopia.**

| Authors | Study Design | Study Year | GL* in Ethiopia | Region | SS | E* | NE* | P* |
|---|---|---|---|---|---|---|---|---|
| Abera*et al.*, [22] | **Cross-sectional** | 2018 | Northwest | Amhara | 300 | 65 | 235 | 22 |
| Alemayehu and Asrat, [11] | | 2014 | Northwest | Amhara | 384 | 44 | 340 | 11 |
| Aragaw and Tilahun, [28] | | 2014 | Northeast | Afar | 369 | 61 | 308 | 17 |
| Chanie*et al.* [29] | | 2013 | Northwest | Amhara | 270 | 37 | 233 | 14 |
| Defersha and Belete, [23] | | 2017 | Northwest | Amhara | 304 | 61 | 243 | 20 |
| Gebremeskel*et al.* [30] | | 2011 | Northwest | Tigray | 389 | 69 | 320 | 18 |
| Habtamu and Wolde, [12] | | 2010 | Northwest | Amhara | 687 | 256 | 431 | 37 |
| Kebede*et al.* [20] | | 2008 | SothWest | Oromia | 384 | 22 | 362 | 5.8 |
| Kerie and Seyoum, [10] | | 2016 | Northwest | Amhara | 532 | 143 | 389 | 26.8 |
| Melkamu, [26] | | 2012 | Northwest | Amhara | 400 | 73 | 327 | 18.2 |
| Merawe*et al.* [31] | | 2014 | Southern | SNNPR | 252 | 152 | 100 | 60.3 |
| Setargew*et al.* [18] | | 2010 | Northwest | Amhara | 167 | 17 | 150 | 10 |
| Shiferaw and Deressa [33] | | 2015 | Western | Oromia | 384 | 88 | 296 | 23 |
| Tsega and Derso[ | | 2013 | Northwest | Amhara | 384 | 69 | 315 | 18 |
| Wudeneh, [24] | | 2010 | Northwest | Amhara | 442 | 72 | 370 | 16.3 |
| Yeneneh*et al.* [21] | | 2017 | Northwest | Amhara | 384 | 232 | 152 | 60.4 |
| Yihuni*et al.* [27] | | 2015 | Northwest | Amhara | 360 | 80 | 280 | 22.2 |
| Mohammed *et al.* [32] | | 2015 | Southwest | Oromia | 179 | 30 | 149 | 16.8 |
| Aylate*et al.* [13] | | 2016 | Northeast | Tigray | 405 | 75 | 330 | 18.5 |
| Bertualem *et al.* [34] | | 2012 | Northeast | Afar | 400 | 183 | 217 | 45.7 |

**Key**: GL* = Geographical Locations, E* = Event, NE* = Non Event, P* = Prevalence in %, SS = Sample size

### Data synthesis and statistical analysis

To determine the weighted-mean estimate of prevalence for bovine schistosomiasis across the included studies, a random-effects meta-analysis model with a 95% confidence level was used to pool the studies. STATA version 17 and Comprehensive Meta-Analysis software version 3 were used to conduct the overall meta-analysis (overall effect size, heterogeneity, and weight of each study), subgroup analysis, and pooled prevalence odd ratio (Biostat, Englewood, USA). The Cochran's Q test (reported as the p-value) and inverse variance index ($I^2$) were used to assess study heterogeneity, which describes the percentage of observed total variation between studies that is due to heterogeneity rather than chance. As explained by Higgins and Thompson [16], the $I^2$ index was estimated to represent low, moderate, and high heterogeneity, if this corresponds to $I^2$ values of 25%, 50%, and 75%, respectively. The *p*–values (p<0.10) are low suggesting that heterogeneity is present across studies. The presence of heterogeneity between studies was assessed using a forest plot diagram and a Galbraith plot. The forest plot diagram displayed weights, effect sizes, and 95% confidence intervals for individual studies (CLs). Similarly, subgroup analyses for the prevalence of bovine schistosomiasis were conducted with 'study year', study region, and sample size to determine the specific variability between studies.

Publication bias was visualized using funnel plot diagramsand Egger's [17]. Egger's regression test is used to test the funnel-plot symmetry.

## Result

### Search results

As viewed in Fig 1, a total of 345 articles were browsed through different electronic databases and with other methods. A total of 186 articles were removed through duplicate, marked as ineligible and other reasons. A total of 79 articles were eliminated through title and abstract screening. Eighty (80) articles reports sought for retrieval and 42 evaluated for eligibility. Finally, only 20 full-text articles were included for qualitative and quantitative synthesis.

### Characteristics of included studies

Published studies on schistosomiasis infection in cattle in Ethiopia were included in this systematic review. It includes a total of twenty relevant studies for quantitative and qualitative verification. A cross-sectional study design was used in all included studies. The included studies for this systematic review and meta-analysis were conducted in different parts of Ethiopia between 2008 and 2018. The included studies in each region were 12 (60%) in Amhara, 3(15%) in Oromia, 2(10%) in Tigray, 2(10%) in Afar, and 1(5%) in SNNPR. In this systematic review, 167 [18] cattle served as the minimum sample size and 687 cattle were used as the maximum sample size [12]. All included studies used coprological examination with sedimentation technique for the recovery of the schistosomiasis eggs (standardized laboratory procedures) as described by Hansen and Perry, [19]. To evaluate the prevalence rate of bovine schistosomiasis in Ethiopia, 7,378 cattle were used in this case. The prevalence of bovine schistosomiasis ranged between 5.8 and 60.4% [20,21]. The detailed characteristics of the included studies are presented in (Table 1).

### Regional distribution of studies

A total of 20 studies were distributed across Ethiopia's five regions (Amhar, Oromia, Afar, Tigray, and SNNPR). The majority of the included studies were from Amhara regional state in northwest Ethiopia, with three studies from Oromia, two from Tigray, and two from Afar

Region (Table 1). The Amhara region had the highest prevalence (60.4%) of bovine schistosomiasis, while the Oromia Region had the lowest (5.8%).

## Meta-analysis and subgroup analysis

On the Forest plot, significant heterogeneity was observed between the included studies bovine schistosomiasis studies in cattle ($I^2$ = 88.04%; p<0.001) and the pooled effect size was 24% (95% CI: 17% to 31%) (Fig 2). Similarly, heterogeneity and outliers are examined using the Galbraith plot, which plots the expectation of studies within 95% of the range bounded by the two confidence interval lines (Fig 3).

As shown above, in Fig 3, the Galbraith plot confirmed the presence of heterogeneity between studies as almost 95% of studies were not included within the confidence interval. More than eight of 20 studies were not confined to the 95% Confidence interval of the plot.

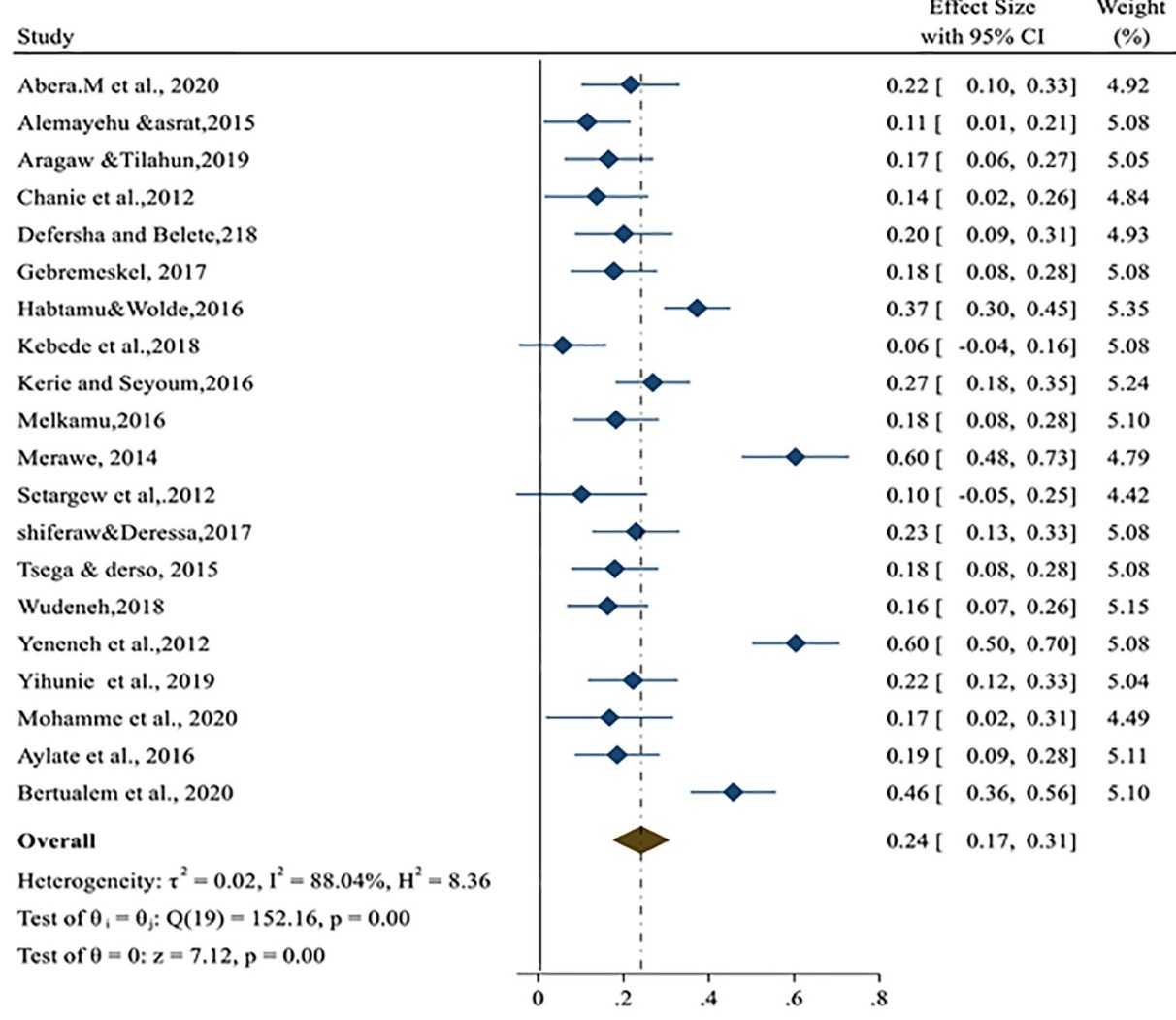

Fig 2. Forest plot for the prevalence of bovine schistosomiasis in Ethiopia.

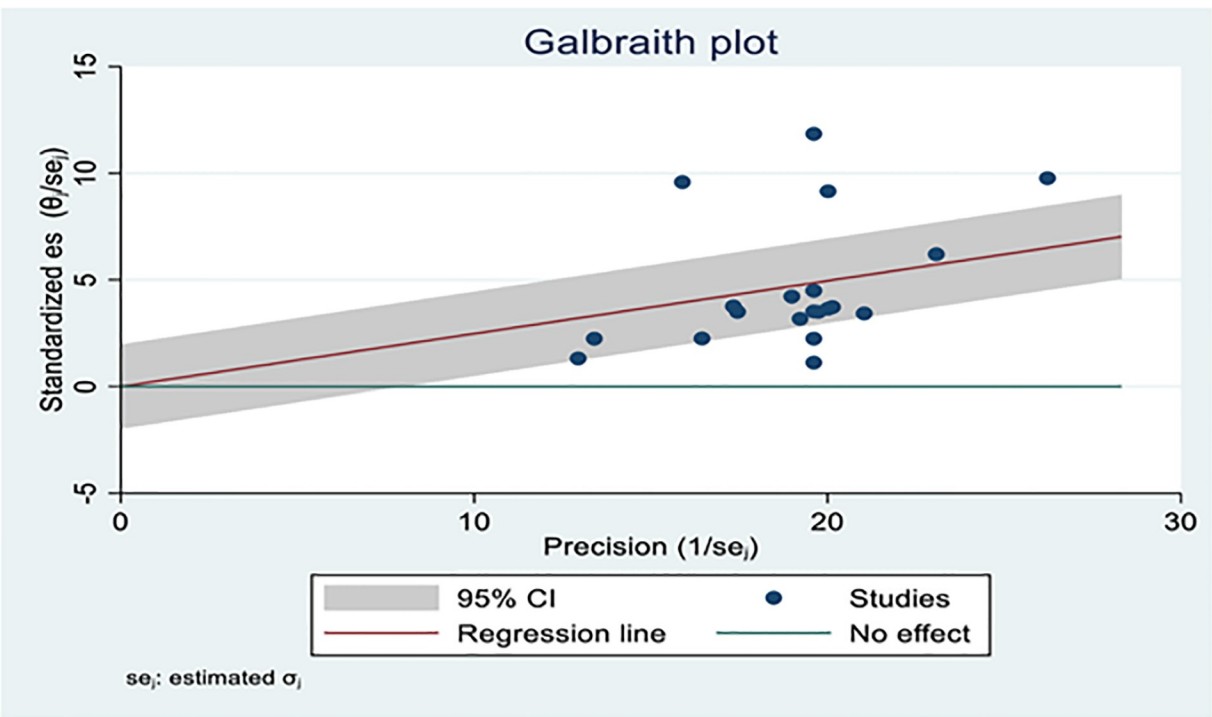

**Fig 3. Galbraith plot portrayed the heterogeneity of included studies for the prevalence of Bovine Schistosomiasis.**

Given the high level of heterogeneity, a subgroup analysis using study year, sample size, and study location (region) was performed to assess the source of heterogeneity and to estimate the effect size between studies using a mixed effects model (Table 2). All study years were divided into three groups (2008–2011) study year category (1), (2012–2014) study year category (2) and (2015–2028) study year category (3) (Fig 4). Subgroup analysis showed that the 2012–2014 study year category had the highest heterogeneity ($I^2$ = 91.78%; P<0.001) with a pooled prevalence of 26%, followed by 2015–2018 ($I^2$ = 86, 07%; p<0.001) with a pooled prevalence of 18% and 2008–2011 ($I^2$ = 84.14%; p<0.001) with an overall prevalence of 26%. There is a statistically significant subgroup effect overall according to a test for subgroup differences (Q = 1.40, p = 0.50).

Subanalysis by region revealed significant heterogeneity between studies, and testing for subgroup differences suggests that there is a statistically significant subgroup effect (Q = 38.8; DF = 4; p<0.001). Subgroup analysis of bovine schistosomiasis by region had shown the highest heterogeneity ($I^2$ = 93.91; Q = 16.41, p < 0.01) (Fig 5) in Afar region.

Based on regional subgroup, the highest overall prevalence of bovine shistosomiasis was reported in SNNPR (60%), followed by Afara (32%), Amhara (23%), Tigray (18%) and Oromia (15%). Analysis by sample size (n = 170–370), (n = 384), and (n = 400–700) also showed significant heterogeneity between studies. Sub-analysis by sample size category, the largest effect size was found in the range 170 to 370 at 28% with $I^2$ = 91.53%; Q = 87.42; p<0.01 (Fig 6). Tests for subgroup differences indicate that there is a statistically significant subgroup effect (Q = 152; DF = 2; p<0.001) between studies to pool estimates of schistosomiasis in cattle. The pooled prevalence of schistosomiasis in cattle in the study year category (n = 384) and (n = 400–700) was 15% and 26%, respectively.

**Table 2. Summary of heterogeneity and subgroup analysis based on study year, sample size and study regions.**

| Subgroup | Prevalence (95%CI) | Weight (%) | $I^2$ | $tau^2$ | Heterogeneity test *p-value* | ES* | Number of studies |
|---|---|---|---|---|---|---|---|
| **Study year** | | | | | | | |
| 2008–2011 | 18(0.07, 0.29) | 25.07 | 84.12% | 0.01 | 0.00 | 0.18 | 5 |
| 2012–2014 | 26 (0.12, 0.40) | 35.04 | 91.78% | 0.03 | 0.00 | 0.26 | 7 |
| 2015–2018 | 26 (0.16, 0.36) | 39.89 | 86.07% | 0.02 | 0.00 | 0.26 | 8 |
| Overall | 24 (0.17, 0.31) | 100.00 | 88.04% | 0.02 | 0.00 | 0.24 | 20 |
| **Sample Size** | | | | | | | |
| N = 170–370 | 28(0.13, 0.42) | 38.16 | 91.52% | 0.04 | 0.00 | 0.28 | 8 |
| N = 384 | 15 (0.07, 0.22) | 20.32 | 53.85% | 0.02 | 0.00 | 0.15 | 4 |
| N = 400–700 | 26 (0.18, 0.33) | 41.17 | 80.63% | 0.01 | 0.00 | 0.26 | 8 |
| Overall | 24 (0.17, 0.31) | 100.00 | 88.04% | 0.02 | 0.00 | 0.24 | 20 |
| **Region** | | | | | | | |
| Amhara | 23(0.15–0.31) | 60.2 | 86.31 | 0.00 | 0.00 | 0.23 | 12 |
| Oromia | 15(0.4–0.26) | 14.65 | 63.26 | 0.001 | 0.06 | 0.15 | 3 |
| Afar | 32(0.03–0.6) | 10.15 | 93.9 | 0.04 | 0.00 | 0.32 | 2 |
| Tigray | 18(0.11–0.25) | 10.19 | 0.001 | 0.00 | 0.91 | 0.18 | 2 |
| SNNPR | 60(0.48–0.73) | 4.79 | - | - | - | 0.60 | 1 |
| overall | 24(0.17–0.31) | 100 | 88.04 | 0.02 | 0.00 | 0.24 | 20 |

ES* = effect size.

**Sub-group analysis based on various risk factors.** Subgrou analysis was performed based on the persumed risk factors such as body condition, sex, breed type and age of the animals as summarized in (Table 3).

*Body conditon*: sub-analysis of all ten identified studies on body condition was performed, the overall pooled proportion was 20% (95%Cl; 14–25%) with $I^2$ = 61.50%; p = 0.00). The greatest inter-study variability was observed in group (1) or poor body conditions of the animals ($I^2$ = 76.87, $tau^2$ = 0.03; P = 0.00). No significance heterogeneity ($I^2$ = 0.00; P = 0.81) was observed in medium (2) and good body (3) condition animals, respectively. In general, the highest pooled prevalence of schistosomiasis in cattle for the subgroup analysis was in poor (33%), followed by medium (16%) and good (9%) body condition (S1 Fig).

*Sex*: Subgroup analysis was performed based on gender (1 = male, 2 = female) with a total of 18 studies. The overall effect size was 20% (95% Cl; 14–26%) and the heterogeneity was ($I^2$ = 82.16%; p = 0.00). The higher variability was observed in females ($I^2$ = 83.44%, p = 0.00) compared to males ($I^2$ = 81.28%; p = 0.00) (S2 Fig). The pooled prevalence of schistosomiasis in cattle was 20% and 21% in males and females, respectively.

Based on nine identified studies, a sub-analysis was performed to examine the source of heterogeneity within and between studies. There was no clear evidence of inter-study heterogeneity for studies reporting the breed effect on the prevalence of schistosomiasis in cattle ($I^2$ = 9.29%, p = 0.28) (S3 Fig); the pooled effect estimate was 18% (95% CL: 14–21%). The pooled prevalence of bovine schistosomiasis in local and cross-breed cattle was 17% and 18%, respectively.

*Age*: Eight studies were used for subgroup-analysis to determine the variability across studies and within studies. So, the result was revealed significant heterogeneity (Q = 114.84, $I^2$ = 78.49%, p = 0.00) (S4 Fig). The overall effect of bovine schistosomiasis was found 23% regarding to the age of the animals. The sub-group analysis of bovine schistosomiasis by age category

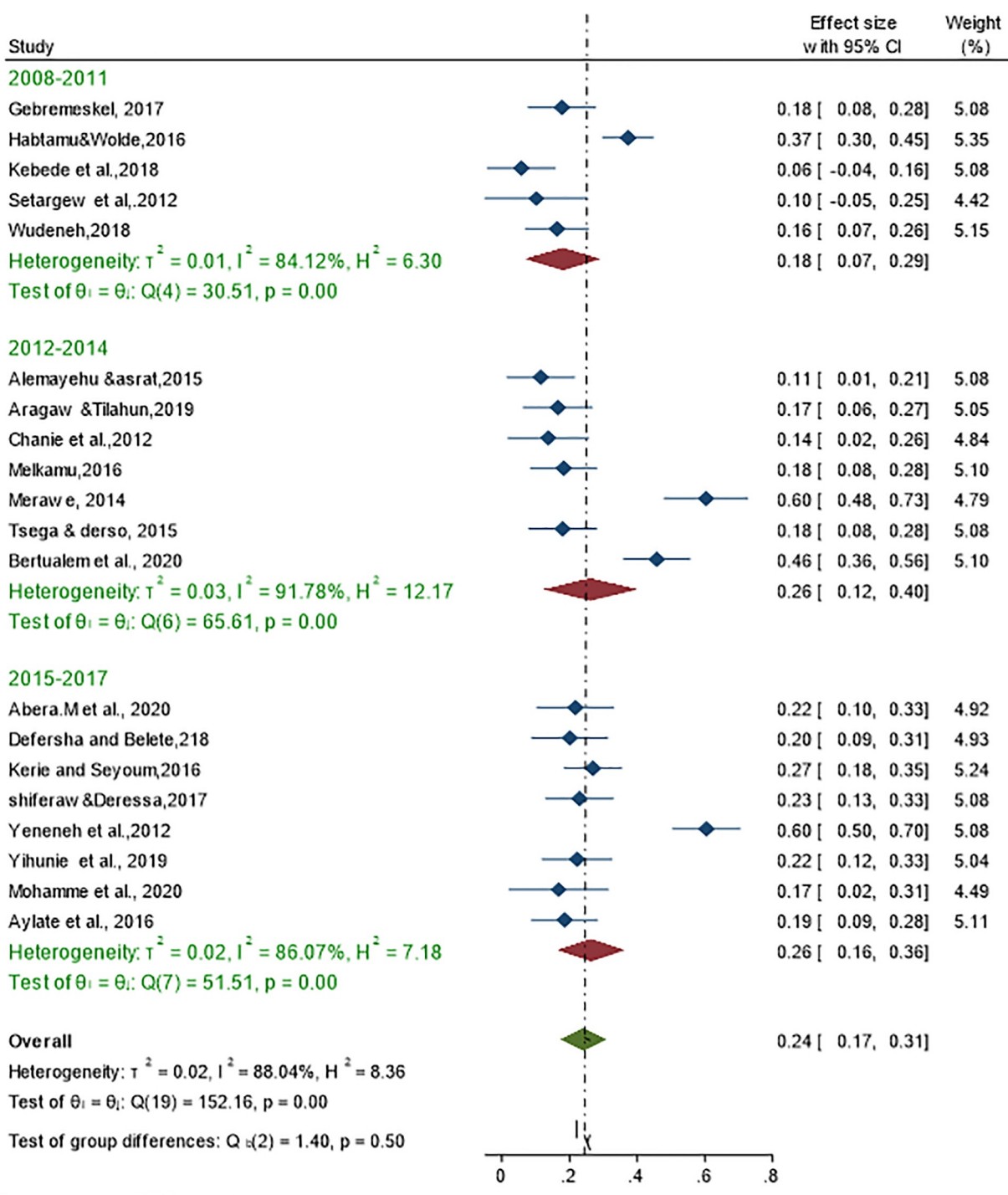

**Fig 4. Forest plot for subgroup analysis of the prevalence of bovine schistosomiasis with study year.**

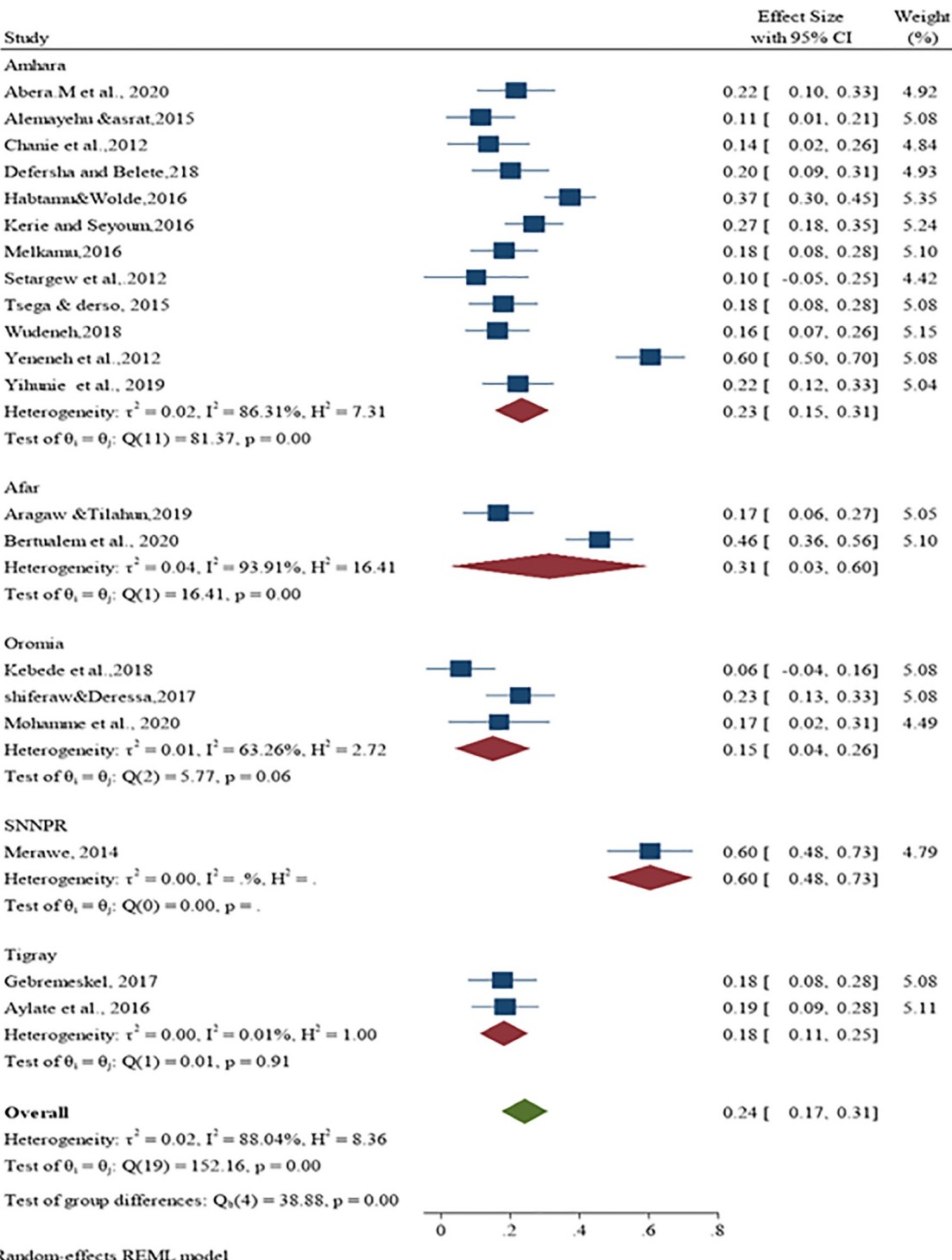

**Fig 5. Subgroup analysis of the prevalence of bovine schistosomiasis based on the regional distribution of the studies.**

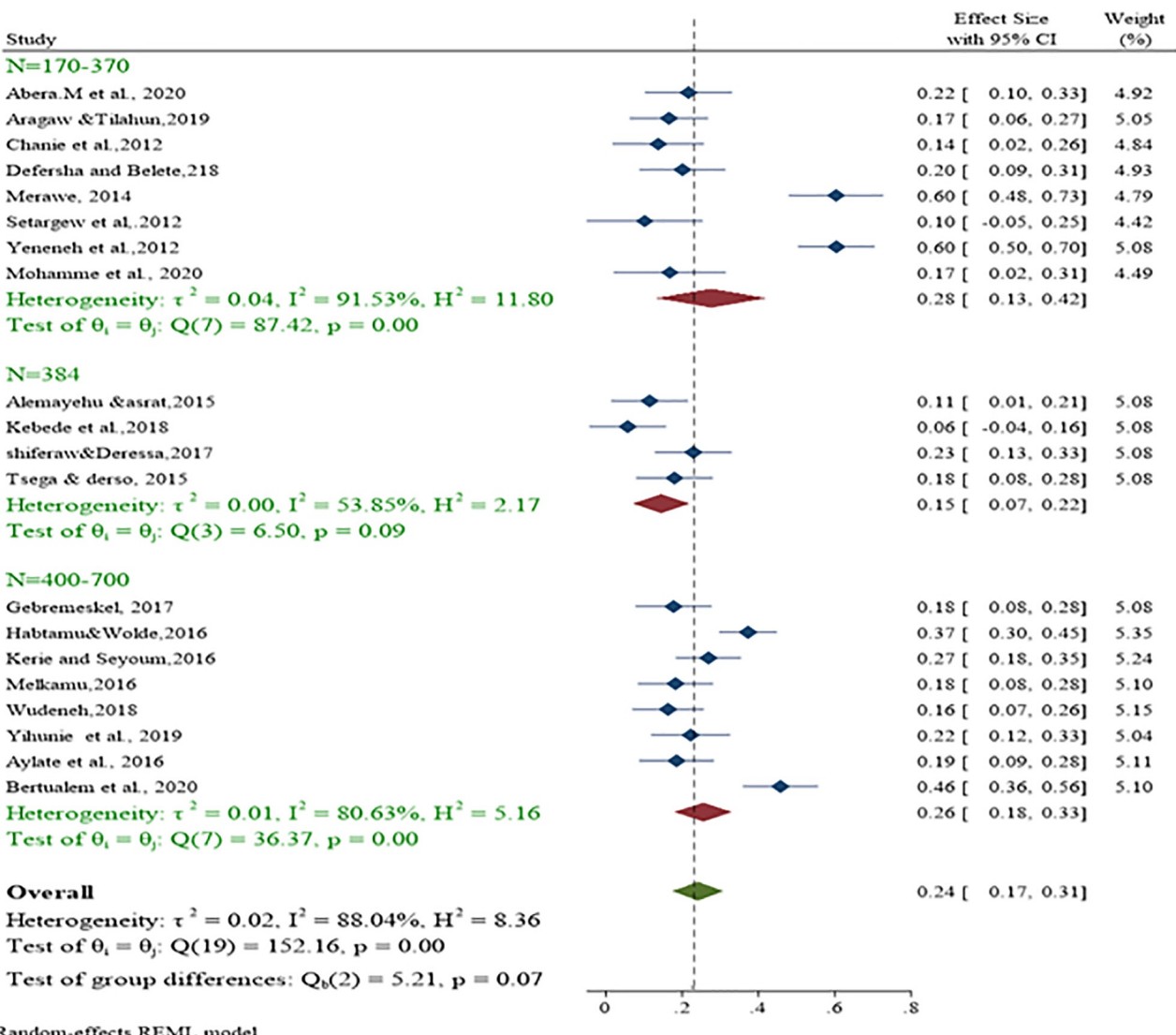

**Fig 6. Statistical summary and subgroup analysis based on sample size category.**

had shown the highest pooled prevalence, 28% (13–43%) for 2–5 age categories, followed 22% for > years, 8% for < 2 year category.

## Bias assessment

The funnel plot (Fig 7) revealed that there was no asymmetrical distribution of studies and all studies are under a 95% confidence interval which means smaller studies do not tend to be missed.

There was no publication bias because Egger's test regreses the standardized effect on their precision. As shown in Fig 8, the Egger's test (ET) plot revealed that there was no statistically significant publication bias in estimating the prevalence of *Schistosoma* infection in cattle (Egger's test: b = -2.58, 95% CI; 0.17–0.31; p = 0.5086.

**Table 3. Summarizes the heterogeneity and effect sizes between hypothesized risk factors and bovine schistosomiasis prevalence.**

| Risk factors | Group | df | Q | P | tau$^2$ | I$^2$ (%) | H$^2$ | ES | N |
|---|---|---|---|---|---|---|---|---|---|
| **Body condition** | Poor | 9 | 28.95 | 0.00 | 0.03 | 76.87 | 4.32 | 0.33 | 10 |
| | Medium | 9 | 5.27 | 0.81 | 0.00 | 0.00 | 1.00 | 0.16 | |
| | Good | 9 | 3.16 | 0.95 | 0.00 | 0.00 | 1.00 | 0.09 | |
| Overall | | 29 | 75.85 | 0.00 | 0.015 | 61.15 | 1.00 | 0.20 | |
| **Sex** | Male | 17 | 115.76 | 0.00 | 0.030 | 81.28 | 5.34 | 0.20 | 18 |
| | Female | 17 | 108.85 | 0.00 | 0.026 | 83.44 | 6.04 | 0.21 | |
| overall | | 35 | 224.64 | 0.00 | 0.027 | 82.16 | 5.61 | 0.20 | |
| **Breed** | Local | 8 | 9.85 | 0.276 | 0.001 | 14.25 | 1.17 | 0.18 | 9 |
| | cross | 8 | 9.84 | 0.277 | 0.004 | 22.27 | 1.29 | 0.17 | |
| Overall | | 17 | 19.84 | 0.283 | 0.001 | 9.29 | 1.10 | 0.18 | |
| **Age** | <2 | 7 | 8.06 | 0.00 | 0.005 | 28.77 | 1.40 | 0.18 | 8 |
| | 2–5 | 7 | 38.67 | 0.327 | 0.036 | 80.82 | 5.21 | 0.28 | |
| | >5 | 7 | 63.0 | 0.00 | 0.064 | 88.71 | 9.86 | 0.22 | |
| Overall | | 23 | 114.84 | 0.00 | 0.033 | 78.49 | 4.65 | 0.23 | |
| **Managment** | Extensive | 3 | 3.9 | 0.29 | 0.00 | 25.19 | 1.34 | 0.20 | |
| | Semi -intensive | 3 | 7.51 | 0.06 | 0.03 | 59.93 | 2.50 | 0.27 | 4 |
| Pooled effect | | 6 | 12.12 | 0.57 | 0.0 | 34.2 | 1.52 | 0.21 | |

## Risk factor studies

Several risk factors have been documented in previous reports in Ethiopia and are presented in (Table 4) and the risk factors considered to affect the occurrence of Schistosoma infection in cattle are geographical locations, age, sex, body condition score (poor, medium or good), breed (cross or local), farm management practices (intensive, semi-intensive or extensive).

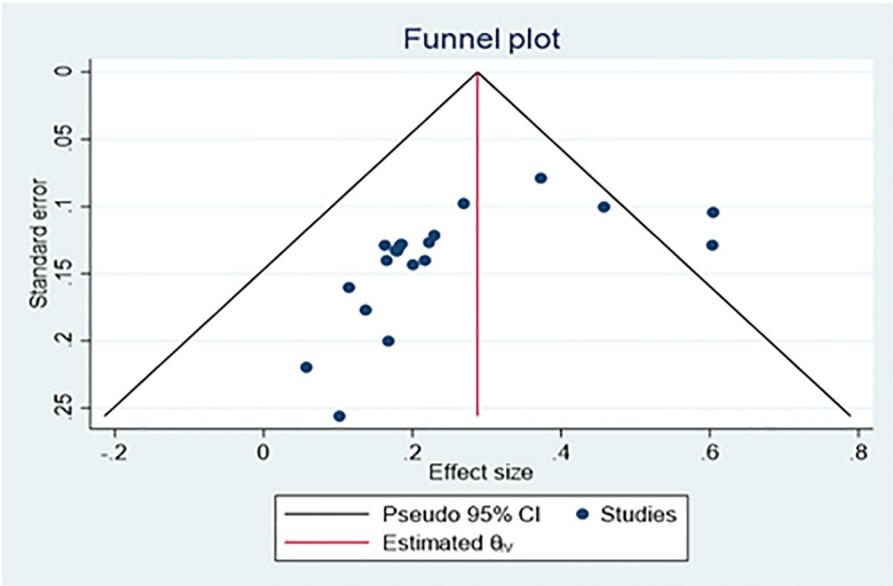

**Fig 7. Funnel plot with pseudo 95% confidence limits of pooled prevalence of bovine schistosomiasis in a different region of Ethiopia.**

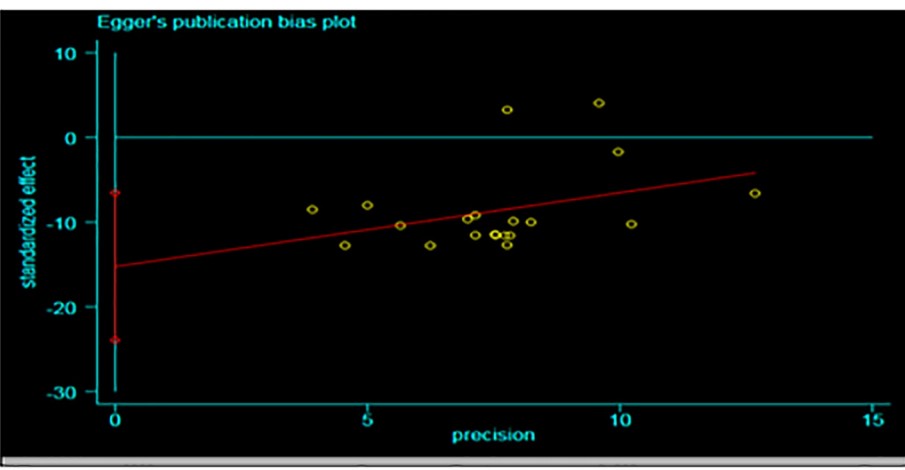

**Fig 8. Egger's plot for publication bias assessment.**

**The association of body condition with bovine schistosomiasis.** Out of twenty included studies, ten studies addressed the association between body conditions and the occurrence of schistosomiasis in cattle in Ethiopia. Eight studies found a statistically significant association between body condition and the prevalence ofschistosomiaisis in cattle, while the remaining two studies found that body condition did not affect the occurrence of this trematode infection in cattle. Therefore, the prevalence of schistosomiasis in cattle was higher in poor body condition animals [13,22,23,26,28,33], followed by medium body condition animals [10] (Table 4). On the other hand, [20,30] found no significant association between the prevalence of schistosomisis and the body condition of the animals which was 24% (95% CI; 22–26%; p = 0.79) and 57% (95%CI 38–85%; p = 0.073), respectively.

**Hetrogeneity**; A subse meta-analysis of all ten identified studies was performed, the overall mean effect size was 20% (95%Cl; 14–25%) and heterogeneity ($I^2$ = 61.50%; p = 0.00). The highest variability between studies was seen in poor body conditions of the animals ($I^2$ = 76.87, $tau^2$ = 0.03; p = 0.00). In case of medium and good body condition no significance heterogeneity ($I^2$ = 0.00; p = 0.81 and $I^2$ = 0.00; p = 0.958) was observed, respectively.

**Table 4. Associated risk factors with bovine schistosomiasis in Ethiopia.**

| Risk factors | Statistically significant studies with risk factors ($p<0.05$) | Not statistically significant risk factors (p>0.05) | No of studies |
|---|---|---|---|
| Age | >5 years by Abera*et al.* [22], 2<x < 5 years by Defersha and Belete [23], 2–5 years by Tsega&derso [24], 2–5 years by Wudeneh [25] | Alemayehu and Asrat, [11], Setargew*et al.* [18], Melkamu [26], Yihunie *et al.* [27] | 8 |
| Sex | Male byAragaw and Tilahun;Wudeneh, [25,28] | Abera *et al.* [22], Alemayehu&asrat [11], Aylate*et al.* [13], Chanie*et al.* [29], Gebremeskel [30], Defersha and Belete [23], Kebede et al. [20], Kerieand Seyoum [10],Habtamu&Wolde [12],Melkamu [26], (Merawe [31], Mohamme*et al.* [32], Tsega&derso [24], shiferaw&Deressa [33], Yihunie*et al.* [27], Setargew*et al.* [18] | 18 |
| Body condition | Poor by Abera*et al.* [13,22–24,26,28,32], poor and medium by Kerie and Seyoum [10] | Gebremeskel [30], Kebede*et al.* [20] | 10 |
| Breed | cross by Abera*et al.* [22], Local by Aragaw&Tilahun [28] and by Habtamu &Wolde [12] | Alemayehu&asrat [11], Defersha and Belet [23], (Tsega&derso [24], Wudeneh [25],Melkamu [26], Yihunie*et al.* [27] | 9 |
| Management system | Extensive by Abera*et a.* [22]Semi-intensive by Aylate*et al.* [13] | Defersha and Belete [23], Melkamu [26] | 4 |

a)

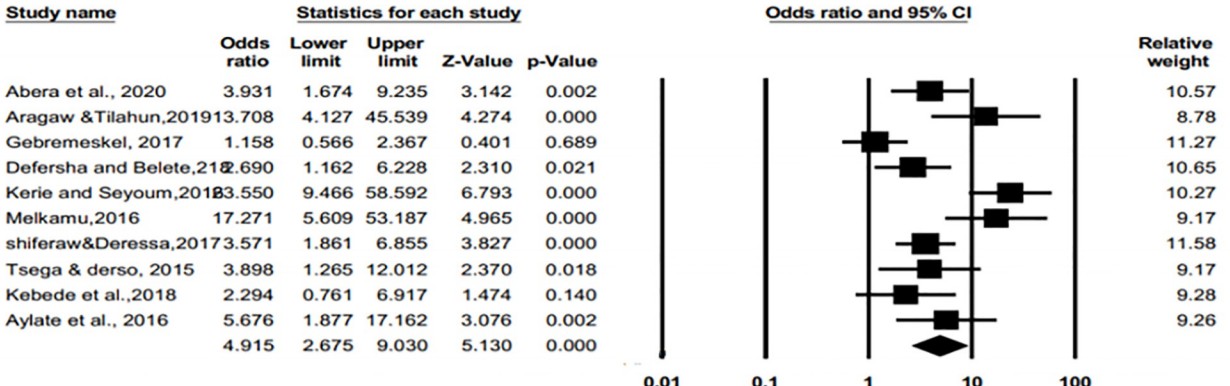

b)

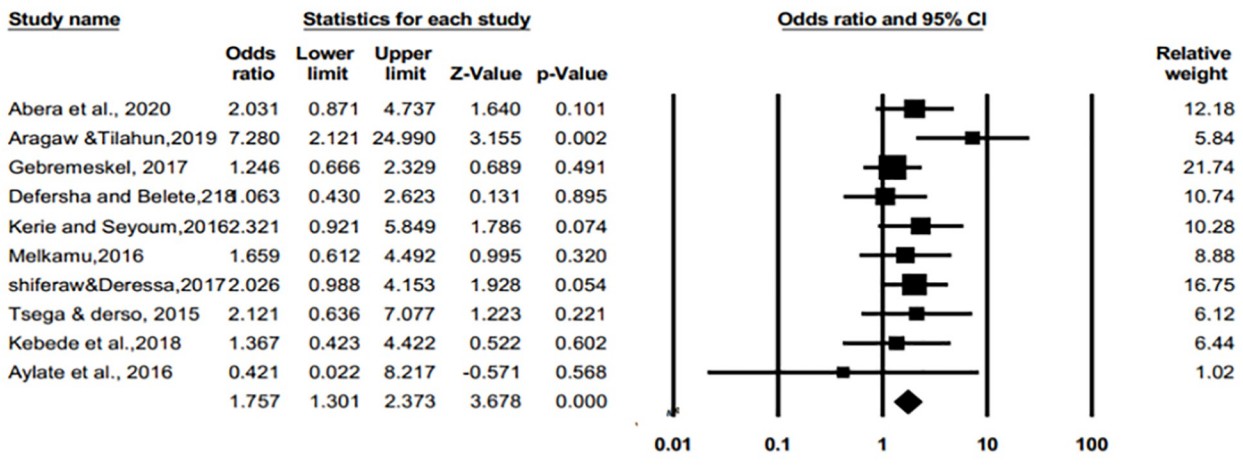

**Fig 9. (a) and (b) Forest plots of odds ratio for the association of poor vs. good body condition (a); medium vs. good body condition (b) with the prevalence of bovine schistosomiasis.**

In this meta-analysis, the pooled odds ratio showed that poor body condition was statistically associated with the prevalence of bovine schistosomiasis (OR: 4.915, 95% CI: 2.675 to 9.030; p = 0.00) (Fig 9a) showed that animals in poor body condition had a 4.915times higher risk of bovine schistosomisis than animals in good body condition. Similarly, medium body condition was positively associated with the prevalence of bovine schistosomiasis (OR: 1.757, 95%CI: 1.301 to 2.373; p = 0.00) (Fig 9b), indicating that animals in medium body condition were 1.757 times more likely to be infected than animals in good body condition.

**The association of sex vs. prevalence of bovine schistosomiasis.** A total of eighteen studies addressed the association between the prevalence of bovine schistosomiasis and the sex of the animals (Table 4). Significant findings were reported in only two studies, there was a positive association between the male gender and the prevalence of bovine schistosomisis [25,28] whereas sixteen studies showed no significant association between the occurrences of bovine schistosomisis with sex.

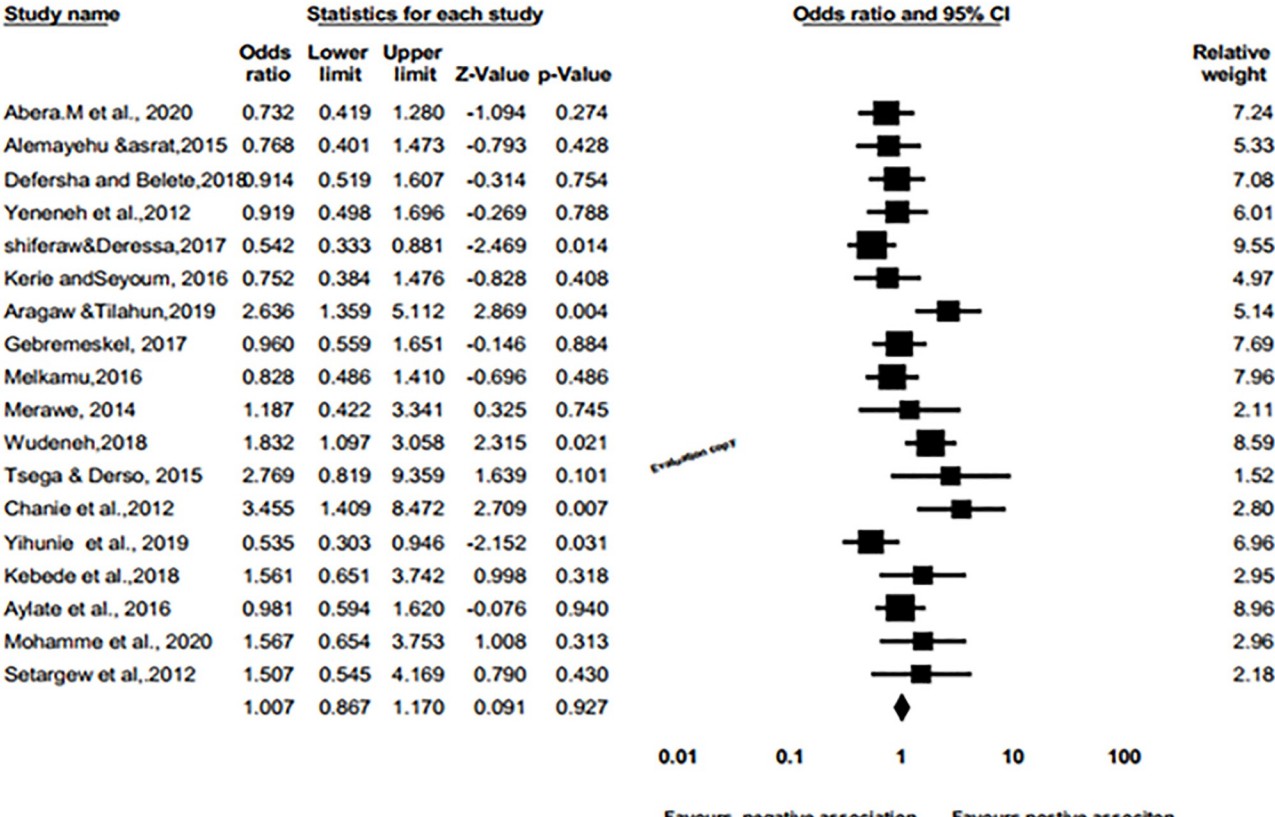

**Fig 10. Forest plots of odds ratio for the association of male vs. female with the prevalence of bovine schistosomiasis.**

Meta-analysis of eighteen included studies was conducted to examine the association between prevalence of bovine schistosomiasis with sex of animals, pooled prevalence odd ratio of sexes were performed in bovine schistosomiasis among cattle population. As shown in Fig 10, the sex of animals was not significant association (pooled OR = 1.007, 95%Cl 0.867 to 1.170; p>0.05) with the prevalence of schistosomiasis.

**Hetrogeneity assesment**: subset meta-analysis was carried out based on gender (1 = male, 2 = female). The overall effect size was 20% (95% Cl; 14 to 26%) and heterogeneity ($I^2$ = 82.16%; p = 0.00). The higher variability was seen in female animals ($I^2$ = 83.44, p = 0.00) compared with male animals ($I^2$ = 81.28%; p = 0.00 (Table 4). So, the current meta-analysis result showed that heterogeneity was statically significant among the studies of bovine schistosomiasis with the sex of the animals.

**Association of breed type with schistosomiasis.** Nine studies addressed the association between breed and the occurrence of schistosomisis among cattle in Ethiopia as depicted Table 4. Three studies found a significant association between breed and the prevalence of bovine schistosomisis [12,22,28]. These studies suggested that bovine schistosomisis was more prevalent in local than the crossbred of cattle. However, six studies did not suggest a significant association between the prevalence of schistosomisis and the breed of the animals [11,21,23–26].

A meta-anlysis of pooled odds ratio examined the association of breed with schistosomiasis in cattle using a random effects model. Hence, the pooled meta-analysis revealed no significant

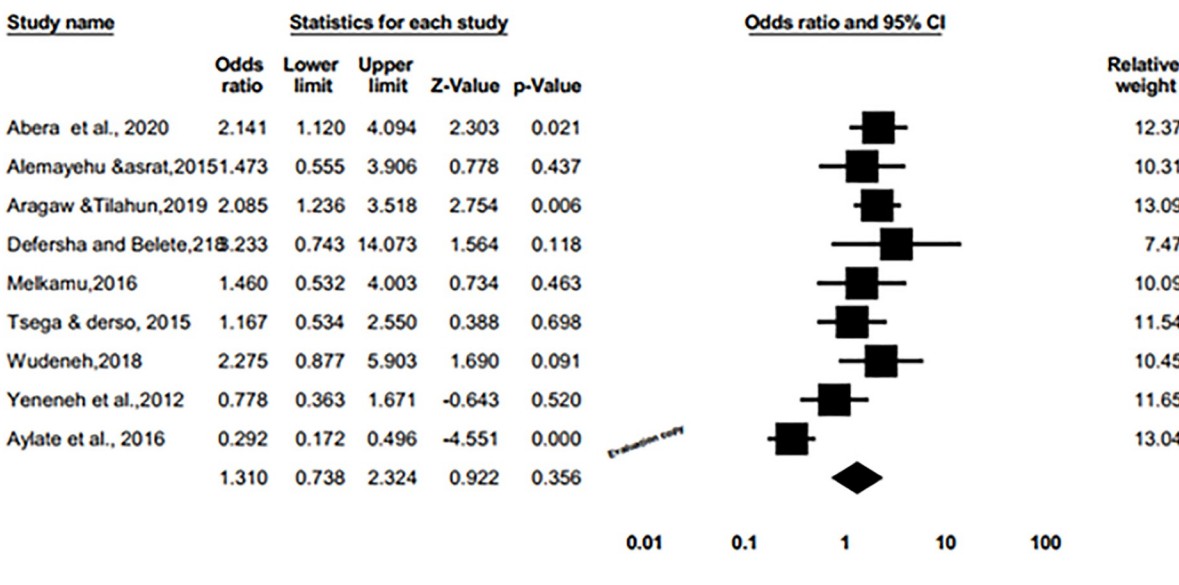

**Fig 11. Forest plots of odds ratio for the association of local vs. cross breed with the prevalence of bovine schistosomiasis.**

difference between the cross and the local of cattle breed with an odds ratio of 1.310 (95%: Cl 0.738 to 2.324; p = 0.356 (Fig 11) with the prevalence of bovine schistosomiasis.

*Hetrogeneity*; based on the nine identified studies, a meta-analysis was conducted to investigate heterogeneity within and between studies using a random effects model. For studies reporting the effect breed on the prevalence of bovine schistosomiasis, there was no strong evidence of between-study heterogeneity ($I^2$ = 9.29%, p = 0.28; (Table 4); the pooled effect estimate was 18% (95%CL: 14 to 21%) in random-effects model.

**Association of age with schistosomiasis.** Eight of the twenty studies examined the relationship between the prevalence of schistosomisis and different ages of the animals in the previous report, discovered a statistically significant association in four of the eight cases by Abera *et* al. [22], (>5 years)& 2–5 years by Tsega&derso; Defersha and Belete; Wudeneh), [23–25] and other four studies reported no a significant association between the prevalence of schistosomisis and the age of the animal) [11,18,21,26]. According to these findings, the prevalence of schistosomiasis was highest in the age category of 2–5 years of age animals followed by animals >5 years of age.

Consequently, meta-analyses of pooled odds ratio of eight studies were conducted to explore the association between different age categories of animals with bovine schistosomiasis. As shown in Fig 12, no significant association was found in three age categories (<2, 2–5, and >5 years) of animals were not shown with the odds ratio 0.0701 (95%; 0.373 to 1.318; Fig 12 (a)), 1.245 (95%; 0.434 to 3.571; Fig 12 (b)) and 1.587 (95%; 0.893 to 2.822; Fig 12 (c)) and p>0.05, respectively. This pooled prevalence is similar to the included studies [11,18,21] and contradict with (>5 years [23] & 2–5 years [23–25].

**Association extensive vs. semi-intensive management system with schistosomiasis.** Four of the identified studies [13,22,23,26] examined the association between the prevalence of bovine schistosomiasis and the management system of the animals. Two of four [13,22] had shown a statically significant association of the management system with bovine

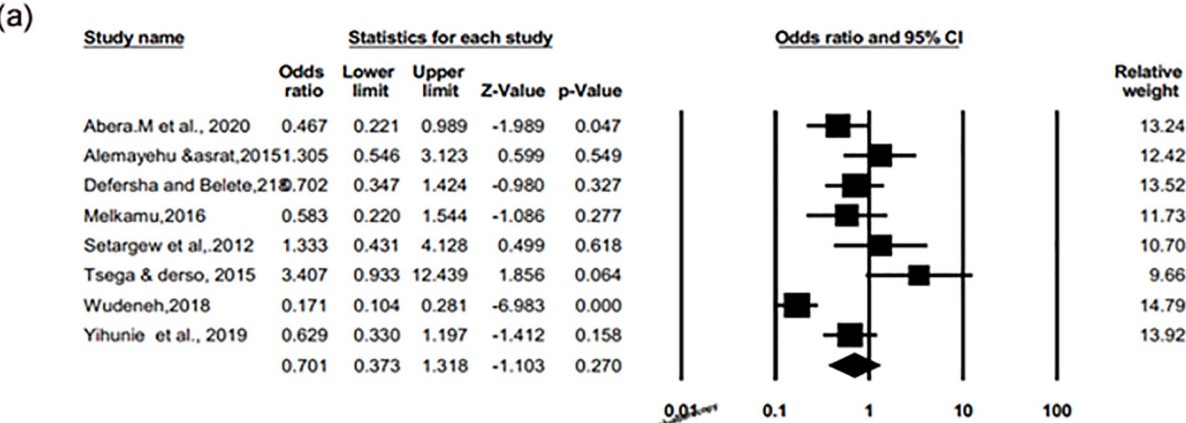

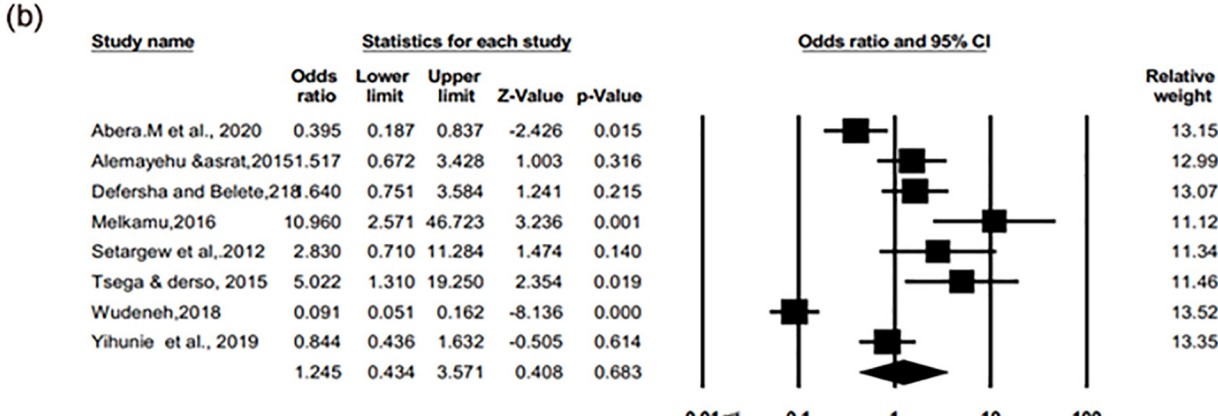

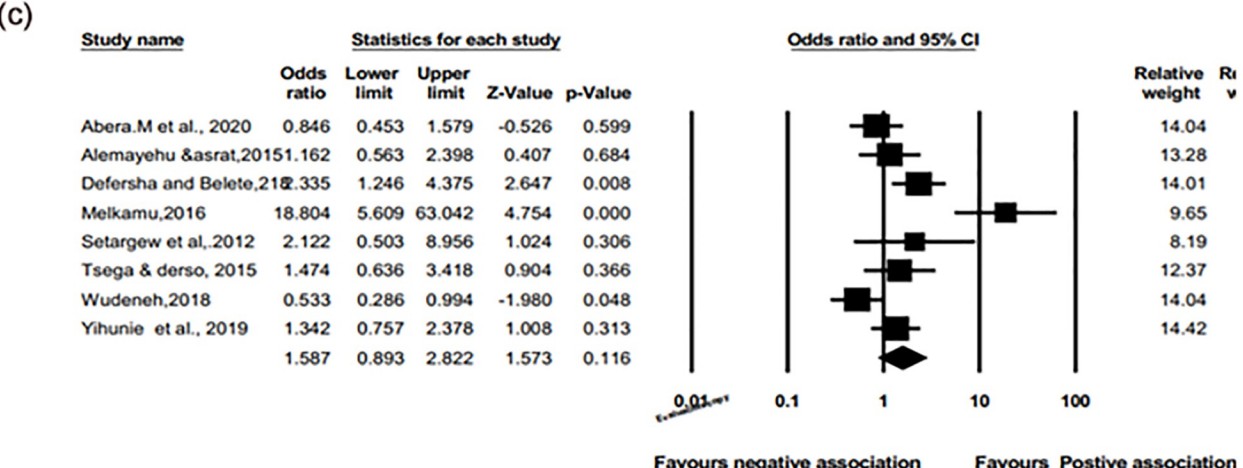

**Fig 12. Odds ratio forest plots for the relationship of age categories with (2 years vs. 2–5) years (a), (<2 years vs.>5) years (b), and (2–5 vs.>5) years(c) categories with the prevalence of bovine schistosomiasis.**

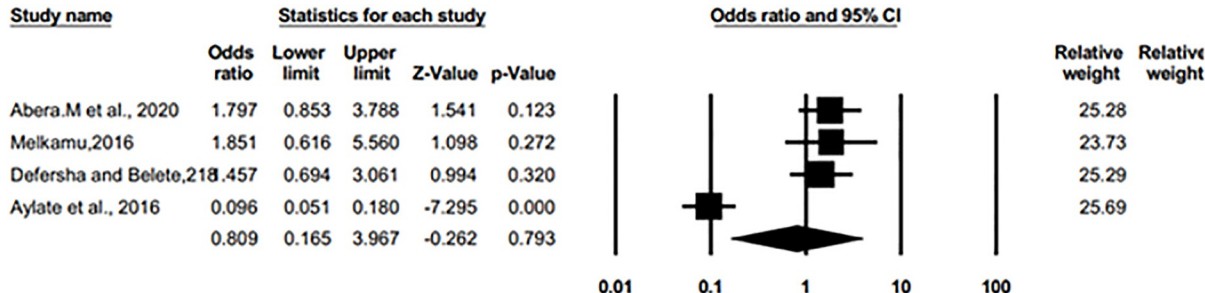

**Fig 13. Pooled odds ratio for a management system with the prevalence of bovine schistosomiasis from random effects meta-analysis of four studies.**

schistosomiasis among cattle. An extensive management system had an effect on the prevalence of bovine schistosomiasis whereas the remaining two studies [23,26] have revealed that the management system had no effect on the prevalence of bovine schistosomiasis.

Therefore, pooled odds ratio analysis was conducted to investigate the relationship between the management system and the occurrence of schistosomiasis infection among cattle. The results in Fig 13 Show that the result of pooled odds ratio showed no significant association (OR = 0.809, 95% CI; 0.165 to 3.967; p = 0.793) across the studies. However, there were a difference prevalence estimation in different management systems, intensive and extensive management systems.

## Discussion

The pooled prevalence of bovine Schistosomiasis in Ethiopia was 24% (95% CI: 17% to 31%). This pooled prevalence was lower than various individual studies conducted in Ethiopia, of which 37% [12], 26.8% [10], 60.3% [31], 60.4% [21] and 45.7% [34]. This low prevalence obtained in this analysis can also be attributed to the lower prevalence in the majority of included individual studies.

Conversely, this pooled prevalence was much higher than 11% [11], 17% [28], 14% [19], 18% [30], 5.8% [20], 18.2% [26], 10% [18], 18% [24], 16.3% [25] and 16.8% [32]. Various research results from various regions of the country showed that the prevalence of schistosomiasis varies greatly between regions. It has been reported that the schistosomaisis infection rate in cattle increases during the rainy season due to an abundance of snails and favorable land conditions that allow the intermediate host to survive compared to the dry season [35]. However, the discrepancy in the prevalence of bovine schistosomiasis between this pooled study and the different individual studies conducted in different period of time and regions of Ethiopia has been implicated with environmental sanitation; the availability of water, and the health of animals, and the socioeconomic level of places are all factors to consider. Furthermore, the disparities noted above could be attributable to methodological variances (i.e.,data analysis, and sampling of study participants).

In light of global concern, the pooled prevalence of bovine schistosomiasis in the current systematic review and meta-analysis finding was also higher than the sero-prevalence report of 6.5% inYola Metropolis, Adamawa State by [36], and 10.0% in Maiduguri Metropolis and Jere Local Government Areas of Borno State [37], in Nigeria. This difference in prevalence may be

partly due to the difference in livestock management practices, diagnostic test used and agricultural land use plans.

The present study revealed that poor body condition has (p<0.05) postive associtioan with the prevalence of bovine schistosomiasis. In the included studies, [32] reported a similar finding. In the present systematic and meta-analysis, cattle having poor body condition (33%), medium body condition (16%) and the prevalence of bovine schistosomiasis was found to be 9% in good body condition.

As a result of this meta-analysis, the infection rate of bovine schistosomiasis was higher in animals with a low body condition than those with a medium or good body condition. Likewse, Fromsa, et al [38], animals with a poor body condition had a higher infection rate of bovine schistosomiasis. This condition could be linked to poor body condition and emaciated animals becoming compromised and susceptible, which could be caused by insufficient nutrient intake, secondary complications, and/or mixed parasite infection [39]. This could be due to low acquired immune status of poor body condition score and weak animals become more suppressed. So, infected animals may take a long time to respond against *schisotoma* infection. This gives a suitable time for the establishment and fertility of parasites in the animals.

Schistosomiasis was found in 20% of male animals and 27% of female animals, respectively. However, no statistically significant difference (P>0.05) was found between the sexes (Table 4). A similar finding was conducted by [11,13,22], the pooled odds ratio of male to female also confirmed that no significant difference (OR = 1.007; p = 0.927) between infection incidence of bovine schistosomiasis and animal gender. This may be due to exposure of both sex categories in similar grazing areas and same watering system and finally equal chance of developing the disease [12].

There was no strong significant association (p>0.05) between the prevalence of bovine Schistosomiasis and the animal management system, according to the quantitative evaluation section of this manuscript. This may be because just a few researches on the relationship between management strategies and schistosomiasis infection rates were included in the studies. Another reason could be this parasite is chronic and it produces a restricted number of eggs. Furthermore, the only method of diagnosis was a microscopic investigation of eggs. Despite no significant difference that links bovine schistosomiasis to the management system, semi-intensive management 27% had a higher infection rate than extensive management 20%. This could be because animals that have been extensively managed animals develop a high level of immunity as a result of repeated natural exposure [40]. Animals reared in a semi-intensive management system, on the other hand, become prone to infection because they lack acquired immunity to the disease.

## Limitation of the study

The study has some limitations. The first limitation could be the investigation (parasitic recovery) method was only a coprological examination. So, the quality of the studies may not be reported as real results, false negative results may be recorded. The second drawback; the greater number of the included articles was obtained from the Amhara regional state whereas the smallest number in SNNPR and no study was obtained from Benishangul Gumuz, Gambella, and Harari. Therefore, this unequal distribution of studies throughout the country may lead to a biased estimation of this finding. The third limitation studies are included in meta-analysis in case of associated risk factors and subgroup analysis was not sufficient in number. The other limitation, even after subgroup analysis, the original data revealed a considerable degree of heterogeneity between trials. As a result, this finding may not necessarily reflect the current condition in Ethiopia's entire region.

## Conclusions

This meta-analysis found that the overall pooled prevalence of bovine schistosomiasis in Ethiopia was high. Thus, this figure shows that schistosomiasis is a widespread and economically important animal disease in Ethiopia. The prevalence discrepancy of the reports can be explained by the differences in location, agroecology, method, and study time. The heterogeneity between different prevalence studies was large. Even in the subgroup analysis, a high degree of variability between regions, year of study, and sample size among the included studies was observed. In the risk factor studies, the occurrence of schistosomiasis in cattle was strongly associated with the body condition of the animals. Based on some empirical evidence and the current pooled prevalence estimate, Ethiopian public health institutions and other responsible bodies should focus on substantially reducing this high level of prevalence as it has public health implications.

## Supporting information

**S1 Fig. Forest plot of subgroup analysis by body condition.** https://drive.google.com/file/d/1xTpipNdNW51PEeLuI0sfTpAcV7wieY9_/view?usp=sharing.
(PDF)

**S2 Fig. Forest plot of subgroup analysis by sex.** https://drive.google.com/file/d/1VTlF4kELV_Td49QMVBK-baYycUybW_ek/view?usp=sharing.
(PDF)

**S3 Fig. Forest plot of subgroup analysis by breed type.** https://drive.google.com/file/d/1ZjTwxBjloIrTbsP7bImgQTZ9t-Q9bzja/view?usp=sharing.
(PDF)

**S4 Fig. Forest plot of subgroup analysis by age.** https://drive.google.com/file/d/1v3yNf9aOhoCMLBHxBDx4bkpoT73vJD17/view?usp=sharing.
(PDF)

**S1 Table. The PRISMA- 2020 Checklist (pdf).** https://docs.google.com/document/d/1A3IO5EguivxXPZAgXgKmO9xzuvaBl9wQ8rXV37dDC-8/edit?usp=sharing.
(DOCX)

**S2 Table. Dataset Microsoft excel.** https://docs.google.com/spreadsheets/d/1umTniOoMi-3DA7eDqGfmmy6vt0j-UOlU3lmCKDmI-e4/edit?usp=sharing.
(XLSX)

**S1 File. Quality assessment tool (AMSTAR-2), highlighted by yellow.** https://drive.google.com/file/d/1CFsTL6TMH8COMize5_PrzgjMGwCTxnnC/view?usp=sharing.
(PDF)

## Acknowledgments

We thank University of Gondar, for providing some of the materials and important pieces of trainings.

## Author Contributions

**Conceptualization:** Melkie Dagnaw, Bihonegn Wodajnew, Tsegaw Fentie, Nigist Bizu.

**Formal analysis:** Melkie Dagnaw, Tsegaw Fentie.

**Methodology:** Melkie Dagnaw, Atsede Solomon, Firdyawukal Abuhay.

**Software:** Tsegaw Fentie.

**Validation:** Melkie Dagnaw, Bihonegn Wodajnew, Atsede Solomon, Firdyawukal Abuhay, Nigist Bizu.

**Writing – original draft:** Melkie Dagnaw, Atsede Solomon, Firdyawukal Abuhay, Nigist Bizu.

**Writing – review & editing:** Melkie Dagnaw, Bihonegn Wodajnew, Atsede Solomon, Firdya-wukal Abuhay, Nigist Bizu, Ahmed Abi.

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
