## [Decision Letter · Decision Letter 0]

20 Jan 2023

PONE-D-22-21332Epidemiology of bovine schistosomiasis and associated risk factors in Ethiopia: A systematic review with meta-analysis of published articles, 2008-2018PLOS ONE

Dear Dr. Fenta,

Thank you for submitting your manuscript to PLOS ONE. After careful consideration, we feel that it has merit but does not fully meet PLOS ONE’s publication criteria as it currently stands. Therefore, we invite you to submit a revised version of the manuscript that addresses the points raised during the review process.

We look forward to receiving your revised manuscript.

Kind regards,

Hudson Alves Pinto, Ph.D

Academic Editor

PLOS ONE

and https://journals.plos.org/plosone/s/file?id=ba62/PLOSOne_formatting_sample_title_authors_affiliations.pdf.

“No”

4. PLOS requires an ORCID iD for the corresponding author in Editorial Manager on papers submitted after December 6th, 2016. Please ensure that you have an ORCID iD and that it is validated in Editorial Manager. To do this, go to ‘Update my Information’ (in the upper left-hand corner of the main menu), and click on the Fetch/Validate link next to the ORCID field. This will take you to the ORCID site and allow you to create a new iD or authenticate a pre-existing iD in Editorial Manager. Please see the following video for instructions on linking an ORCID iD to your Editorial Manager account: https://www.youtube.com/watch?v=_xcclfuvtxQ.

Additional Editor Comments:

Dear Authors,

I am sorry for a long time it took to decide on this MS. It was challenging to find Reviewers for this MS.

The MS presents a potential for publication. However, the Reviewer presented several points that need to be improved before acceptance. I encourage the authors to revise and send an improved version.

Reviewers' comments:

Reviewer's Responses to Questions

**Comments to the Author**

1. Is the manuscript technically sound, and do the data support the conclusions?

Reviewer #1: Yes

2. Has the statistical analysis been performed appropriately and rigorously? 

Reviewer #1: Yes

3. Have the authors made all data underlying the findings in their manuscript fully available?

Reviewer #1: Yes

4. Is the manuscript presented in an intelligible fashion and written in standard English?

Reviewer #1: No

5. Review Comments to the Author

Reviewer #1: The topic is critical, but the authors must work very hard to improve the manuscript. All important comments are included in the attached word and pdf files. I am confident that by addressing all of the issues raised, the manuscript will be significantly improved.

6. PLOS authors have the option to publish the peer review history of their article (what does this mean?). If published, this will include your full peer review and any attached files.

Reviewer #1: No

---

## [Author Response · Author response to Decision Letter 0]

7 Mar 2023

PLoS ONE,

 PLoS ONE Veterinary Science, 

 Date: 26/02/2023 

Rebuttal letter 

Dear respected Academic Editors

 and Reviwers,

Thank you for allowing a resubmission of our manuscript, with an opportunity to address the reviewers’comments. 

Subjec: response to comments and resubmission of revised manuscript entitled “Epidemiology of bovine schistosomiasis and associated risk factors in Ethiopia: A systematic review with meta-analysis of published articles, 2008-2018” on the Journal of PLoS ONE section PLoS ONE Veterinary Science by Melkie Daganaw Fenta (corresponding author) with manuscript ID number of PONE-D-22-21332. The comments provided by Academic editors and reviewrs are thouthgtfull and constructive. We attempted to answer point-by-point the comments, uploaded revised manuscript with track changes and an unmarked version without track changes. This rebuttal follows the format in which the points from the reviewers/editors were left in the original order, in black, and our responses (in green) were then inserted after each of those points.We hope the revised version is now suitable for publication and look forward to hearing from you in due course.

Withregards, 

Melkie Dagnaw et al.

University of Gondar

Response for the academic editor 

Dear Respected Editor,

Thank you for your valuable comments and details on our manuscript.We carefully answered each point and revised the manuscript accordingly. 

Academic Editor Comment#1: please submitting revised manuscripit with rebuttal letter, revised manuscripit with track change and unmrked version

Author’s response: thank you for your kind suggestion.We revised the uploaded revised manuscript with rebuttal letter, revised manuscript with track change and unmarked version

Academic Editor Comment #2: Please ensure that your manuscript meets PLOS ONE's style requirements, including those for file naming. The PLOS ONE style templates can be found at 

Author’s response: We have followed the PLOS ONE’S style and format in the revised manuscript.

Academic editor comment#3: Thank you for stating the following financial disclosure, NO

Author’s response: the author’s recevied no specific funding for this works and describe on the cover letter.

Academic Editor Comment #4: We note that you have stated that you will provide repository information for your data at acceptance. Should your manuscript be accepted for publication, we will hold it until you provide the relevant accession numbers or DOIs necessary to access your data. If you wish to make changes to your Data Availability statement, please describe these changes in your cover letter and we will update your Data Availability statement to reflect the information you provide.

Author’s response: thank you for the sugestion forwarded, Data Availability statement is already changed and describe on the cover letter.

Academic Editor Comment #5: PLOS requires an ORCID iD for the corresponding author in Editorial Manager on papers submitted after December 6th, 2016. Please ensure that you have an ORCID iD and that it is validated in Editorial Manager. To do this, go to ‘Update my Information’ (in the upper left-hand corner of the main menu), and click on the Fetch/Validate link next to the ORCID field. This will take you to the ORCID site and allow you to create a new iD or authenticate a pre-existing iD in Editorial Manager. Please see the following video for instructions on linking an ORCID iD to your Editorial Manager account: https://www.youtube.com/watch?v=_xcclfuvtxQ.

Author’s response: thank you for your suggestion. The corresponding author has ORCID ID, Authenticated

Response to Reviewer 1:

 Thank you for your valuable comments and details on our manuscript. We carefully answered each point and revised the manuscript accordingly. 

Abstract: 

Comment: as the reviewr commented us there are lungustic and subjective related problems in the abstract. The comments forwarded by the reviwer are obiously accepted by the authors because they are rational and reasonable. Especially, we are thankfull the reviewr as providing the golden idea regrding to the modification of objective of the study. We had tried to correct each point as follows.

 Specific Points comments Response Method refernce lines on page 2

The word cow Replaced by cattle Corrected replaced 33

 The word have 

 changed to has Corrected replaced 35

objective of the studies

 Change Corrected, Deleted the previous one 35-37

“The eligible studies criteria for Systematic Reviews and Meta-Analyses statement protocol were used to conduct this review” Unclear Correceted rephrased 37 &38

“reference list databases” List them correceted Listed the datbases 39

 The phrase “To investigate heterogeinity

Not” Add the word “source” Corrected Added, “To investigate the source of heteriogenty” 42

“The prevalence of bovine “schistosomiasis in cattle was shown to be very variable throughout the investigations” unclear corrected rephrased 49&50

In calculating the prevalence of schistosomiasis Remove this phrase corrected removed ---

“To identify the various risk variables related to the prevalence of bovine schistosomiasis, a pooled odd ratio was used” Shift to methodology part of the abstract corrected Shifted to the proper site 45 &46

the type error related to the odd ratio 

 4.915 Or 4.751, which one?, make spell “p” Correceted Selected(4.915) based on the analysed result and made small letter “p” 54

the genus “shcistosoma” 

 Make italics Corrected Overall manuscrpit

Related to conculusion” the phrase “Therefore, the poor body condition of the animals will be warranted”.

 notclear Correceted rephrased 56&57

Related with the last sentence of the abastract“Further, sufficient numbers of prospective studies will be conducted to address other potential risk factors of bovine schistosomiasis in Ethiopia” Is it your recommendation or do you intended to carry out studies Corrected This our recommendation, so corrected through rephrased 58&59

Introduction

 Comment: as per the reviewr commented in the introduction part this statement should be paraphrased “Schistosomiasis affects productivity, reproduction performances, and medically important parasitic disease of humans and animals, which dwell in the circulatory system of definitive hosts [2]”.

Answer: Thank you for your valuable comments and suggestions. We kindly paraphrased (page No2, Line No 69-71).

Comment: as the reveiewr sugested that kindly make italic“Schistosoma japonicum parasitizes humans, cats, and mammals in Asia; Schistosoma magrebowiei parasitizes antelopes in Central Africa; and Schistosoma rodhaini parasitizes wild rodents”.

Response: we thank reviewer 1 and apology this great mistake, because genus name always should be italics. Therefore we made italicized (page No 3, Line No76&77).

Comment: as the reviewer 1 suggested that in the introduction part of the manuscrpit, “I think this supposed to be the first paragraph “Schistosomiasis is one of the fifteen neglected tropical diseases (NTD) because it is unusual amongst helminth diseases for two reasons: much of the pathogenesis is due to the eggs (rather than larvae or adults), and the majority of the pathology is caused by the host's immune system (delayed-type hypersensitivity and granulomatous reactions) [6] and it is the second parasitic disease next to malaria in the human case.” 

Response: the authors are fully agreed with the comment forwarded by reviewer.therfore we shifted to the first paragraph (page2, lines 64-69).

 Comments: as the reveiewr recommended that, the author’s names should be mentioned in the place of 10-13 “In Ethiopia, various epidemiological studies on schistosomiasis in cattle were undertaken. The prevalence of Schistosoma bovis has been studied by several authors in different periods and different parts of Ethiopia and a prevalence of 12.3-29% were reported by [10-13]”

Response: we are grateful for suggestion and we incorporated in the improved manuscrpit (page4, line 101).

Comment: the reviewer commnted as rerwite and authors should concentrate the on scintific or other potential applications of the review findings”

 Schistosomiasis is a poorly understood tropical zoonotic trematode infection that has received little attention. It is a usual process for summarizing results from several studies that are being used largely in human medicine. However, summarizing results is to a more limited extent in veterinary medicine. Besides these limitations, a systematic review of all available studies and a pooled estimate result of various individual studies (meta-analysis) have not yet been conducted in Ethiopia. As a result, the goal of this systematic review and meta-analysis was to offer a pooled prevalence of bovine schistosomiasis and related risk factors based on available cross-sectional research in Ethiopia.

Response: we thank the reviewer comment forwarded about the descripition of this paragraph. Authors fully accepted comentes and rerwite the paragraphs as per reviewr suggestion, even we have deleted some part of the pharase and add some important points (page 4, lines 106-112). 

METHOD

 Comment: the reviewr 1 sugests that this sstatment is not clear and need parphrasing“The checklist is used to confirm the inclusion of relevant information from the selected articles based on the underline protocols”

Response: Thank you for the suuggestion. As per suggestion, we have paraphrased (page No 5, line No 123 1nd 124).

Indition to at this part the reveiewr also sugested that, “the main goal was to figure out the overall prevalence of bovine schistosomiasis in Ethiopia, as well as the associated risk factors”, shifted to the introduction.

Answer: We accepted the comment; however, according to Reviewer's recommendation regarding changing the objective part of this manuscript, this part should better be removed.

Search strategy 

 Comment: the reviewr coment on the higligted part,”The literature search and other manuscript preparation were conducted between September 01, 2021, and April 28, 2022

Response: we thank the reviewr1 about this piece of idea highlighted above which was marked by??, previously we have added the other manuscript preparation means, we thought that the per-searching procures and other author discussions regarding to the objectives, formulation review questions and task allocations for each authors as per their skills. However, we accepted this part of the comment and decided to remove. Now the sentence becomes,”The literature search were conducted between September 01, 2021, and April 28, 2022 (page No 5, line 128)

Comment:the reviewr suggested that what do you mean “Snowball searching from retrieved articles” Do you mean looking the references to the articles you identified? 

 Response: okay, we have searched new articles in the reference part of the retrived similar studiese and we have corrected (page No 5, lines 131).

 Comment: What are these manual methods?

 Response: open google internet, libraries magazine, chapters, and books (page 5, line 132)

Study eligiblity

 Comment: the reviewr commented to remov this sentences “The study was conducted according to PRISMA guidelines” because repeated

Response: we thank you for the suggestion; we authors checked this statment, it was present in other part of the manuscripit. Then, we greatfully omited (page 5), it was in the first paraagaraph.

Comment: why time restricted in 2008-2918?

Response: Thank you very much for this great comment. The core concept of systematic review to provide and overview of the specific topic systematic approaching including the identifying all the potential evidences, selecting evidences objectively, assessing evidences critically and synthesizing the evidence reasonably. An important step of this compressive procedure is to find potential evidences. Then, researcher, exclude references in the objective way.that is why the eligible criteria are required in the methodology of the systematic review and Meta analysis. This procedure helps primarily to avoid bias and to provide the reliable outcome. Without reason to exclude articles is not critical procedure of systematic review.

 Therefore, in our case, some of the eligible criteria were

 1. Clear and detailed presentation of results and context 

2. Publication of articles between 2008 and 2018.

Response for why not include studies before 2008 

No clear presentation and lack of published studies before 2008, this due to lack of veterinary professionals, although there were no specialists such as parasitologists, pathologists and microbiologists. Universities, research institutes of the veterinary service have not been developede

Response for why not include later than 2018

Between 2019 and 2022 there was a war in most parts of our country (Ethiopia), especially in Amhara, Tigray and Oromia regions, in those years there was no teaching, no research, b/c universities, research centers were closed. However, peace reigned in some parts of the country and there was some activity. Therefore, in order to remove the bias estimate, we decide to set this time period as the allowable criterion. 

Comment: As per recommendation of he freviewr1, this statment “Articles were selected if they coincide with the above inclusion criteria” should be omit?

 Response: Since this sentence is repeatition and unsessery, we accepted the comment and decide to omit.

Further the reviewr 1 commented on “Finally, these inclusion and exclusion criteria were used for the extraction of data and a meta-analysis (study screening strategy and exclusion reasons) is presented (Fig.1)” is unclear statement.

 Response: We tried to paraphrased and clarifed on page No 6 and lines 157 and 158

Study quality assessment:

The reviewer kindly requested us to omit this statement”Quality assessment is performed precisely by standard tools because meta-analysis considers the highest level of evidence in the first “on Study quality assessment

Response: we thank the reviewer 1 for this suggestion; we assumed previously this statement as the introduction and clarification for quality assessment, however it is not important to put here after the comment provided by the reviewer. Now, we omitted

Comment: There are many tools that are available to use for checking the quality of articles during systematic review and Meta analysis, such as the Joanna Briggs Institute (JBI) critical appraisal checklist. I suggest authors use one of them and report the findings accordingly.

Response: we thank and request to apologize for this great mistake, really we all authors accepts fully and incorporated the quality assessment tool (page No 6, line 169) and remove unnecessary statements 

Data synthesis and statistical analysis

The reviewer 1 recommended us do not make italics “bovine schistosomiasis”as per the recommendation of the reviewer we changed normal font size and remov italics “bovine schistosomiasis” (page 6, line 180).

Reviewer 1 suggested us put the cutting value of p-value “The p –values are low suggesting that heterogeneity is present across studies”, as the requsetd we agreed to incorparte the cutting point of, p-vales which is p<0.10 and suggesting that low heterogeneity is present across studies (page 6, line 189).

Results 

As per the suggestion and recommendation of the reviewers the main modification has made on the this section

PRISMA 

 Comments: “state the reason for excluding n = 25 and n = 79 articles. Why the n = 38 articles were not retrieved? Omit one statement between the two in the included section. I would rather stick with the first.

 Response: As per the recommendation of the reviewer 1 we have included reasons and omited one statement in the included section (page8, Fig1).

 Comment: povide the number and percentage of studies carried out in each of the zones described?

As per requsted kindly to put the number of studeis with percentage by the reviewer 1, we added all the articles with percentage (page 9, lines 262 and 263).

Comments: “Individual studies included in this systematic review had a minimum sample size (cattle) of 167 [20], while the included study had a maximum sample size of 687 [12]”

“The included studies' detailed characteristics are presented in (Table 1)”.

As per the recommendation of the reviewer we have paraphrased these two statements (page 9, lines 263 and 264) and line 269.

Meta-analysis and bias assessment

 The reviewr 1 suggested that this two “Meta-analysis and bias assessment” should be separeted.

As per recommendation of reviewr 1 we separated into two parts. And we combined Meta analysis with subgroup analysis, on page 9, line 278, because the reviewr gave a general comment as “To ensure a smooth flow of information, the results of meta-analysis can be combined with those of subgroup analysis”

 While the “bias assessment” put on page 19 and line 403 separetly

Subgroup analysis

 Comment: the reviewer 1 comment about this statement "Subgroup analysis based on study Region revealed that heterogeneity across studies was greater than sample size or study year”. , what does mean this statement? Is it subgroup of subgroup?

Response: Thanks to the reviewer and credit as this point is very hidden and unclear. We assumed the overall comparison of subgroup heterogeneity across a group; however, we authors have discussed this point. Then we agreed to omit it because it creates confusion and is impossible even to compare within a group

.

 Comment: the reviewr1 commented on the categorzation of the sample size subgroup

 The subgroup analysis by sample size (n=170 - 370), (n=384) and (n=167-687) also revealed substantial heterogeneity between studies unclear categorazation?

Response: We thank you with a big apology for this big mistake. All authors accepted without drought. Now let's clearly categorize this big mistake (page 13, line 316).

The reviewr 1 commented as “Kindly edit the table by replacing this to its appropriate position. Here should be the word subgroup”

 Thank you for that critical point, obviously we accepted. Therefore, as requested by the reviewer, we added and edited words in the correct position (page 19, Table 3)

On this part, the reviewr advised to perform subgroup analysis by risk factors. 

Okay, we've accepted this crucial part of the subgroup analysis to study heterogeneity. We noticed that in the previous manuscript subgroup analyzes were carried out according to different risk factors; However, we have described the analyzed result only by the table (Table 4) without the forest plots and other results in the case of the previous manuscript. We now present the results in both forest plots and in Table 3 on page 19, and the forest plot has been included as supplementary material as S1-Fig. (Page 18, line 366), S2-Fig. (page 18, line 371), S3-Fig (page 18, line 375) and S4 Fig (page 18, line 379).

Discussion

In this part the reveiewr1 suggested regarding with to remove repetation statements, 

 Response: As per the recommendation of the reviewer 1 we have included reasons and omited one statement in the included section (page8, Fig1).

 Comment: povide the number and percentage of studies carried out in each of the zones described?

As per requsted kindly to put the number of studeis with percentage by the reviewer 1, we added all the articles with percentage (page 9, lines 262 and 263).

Comments: “Individual studies included in this systematic review had a minimum sample size (cattle) of 167 [20], while the included study had a maximum sample size of 687 [12]”

“The included studies' detailed characteristics are presented in (Table 1)”.

As per the recommendation of the reviewer we have paraphrased these two statements (page 9, lines 263 and 264) and line 269.

Comment: As a result of this meta-analysis, the infection rate of bovine schistosomiasis was higher in animals with a low body condition than those with a medium or good body condition. Likewse, Fromsa, et al [38], animals with a poor body condition had a higher infection rate of bovine schistosomiasis, how?

Response: thank you for the requested on the reason why question. We incorporated the reason on page 30, lines 613-616).

CONCLUSION

Comment: please omit the statment “It is the first meta-analysis and systematic review attempt to assess the pooled prevalence of bovine schistosomiasis in Ethiopia that we are aware of. The basic guideline for Preferred Reporting Items for Systematic Reviews and Meta-analysis was followed in this systematic review and meta-analysis (PRISMA)”.

Response: we have deleted the statement and from the improved manuscrpit version 

 The overall summery of key comments and sugestions provided by the reviwer and checked by the authors.

S.No Comment and sugestions Response

1 This is critical work in terms of providing evidence-based information for bovine schistosomiasis, particularly in this era of Schistosoma hybridization. As a result, authors should clearly state the purpose of this review. The absence of a systematic review and meta-analysis of bovine schistosomiasis in Ethiopia is not a compelling reason to conduct this review. The reason for this could be to provide information that can be used in the planning and design of Schistosoma interventions in Ethiopia, as the world aims to eliminate schistosomiasis as a public health problem by 2030.

 �

2 The contribution of each author should be provided in each section of this review, particularly during article searching, screening, data extraction, and article quality assessment. This will demonstrate the caliber of their work.

 �

3 There are so many contradictory statements in the review. Authors should paraphrase them to provide clear meanings �

4 There is some statement repetition. Authors should make sure that each statement appears only in its appropriate section �

5 There are so many unnecessary statements in the methodology section. These should be omitted or placed in the introduction or discussion sections.

 �

6 Authors should conduct quality assessments using any of the available standard guidelines and appropriately present the results �

7 The results of the meta-analysis and the bias assessment should be separated by the authors. To ensure a smooth flow of information, the results of meta-analysis can be combined with those of subgroup analysis �

8 To confirm the funnel plot result, I recommend authors present only one test result, choosing between Egger's and Begg's tests. Also, publication bias and quality assessment of all included studies should be done once. No need of the repeated assessments. Studies presented low quality should be omitted from analysis �

9 The data analysis section should clearly state the subgroups (factors) that were involved during subgroup analysis and categorization of eachsubgroup (factor). Furthermore, the categories must be distinct �

10 Subgroup analysis results were limited and poorly presented. The authors did not present the pooled prevalence of bovine schistosomiasis in each category of subgroup with a significant contribution of heterogeneity, nor did they present the amount of heterogeneity contributed by each factor (R2).

 �

11 There are far too many repetitions of the results in the discussion section, rather than discussions of the results. �

12 The conclusion section should be revised by providing statements that generalize all key findings. There is no need to discuss them. �

13 A native English speaker or a professional English language editor should revise the entire manuscript. �

14 Findother comments in the pdf file.

 �

Withregards, 

Melkie Daganw et al.

---

## [Editor Report · Decision Letter 1]

14 Mar 2023

Epidemiology of bovine schistosomiasis and associated risk factors in Ethiopia: A systematic review with meta-analysis of published articles, 2008-2018

PONE-D-22-21332R1

Dear Dr. Fenta,

We’re pleased to inform you that your manuscript has been judged scientifically suitable for publication and will be formally accepted for publication once it meets all outstanding technical requirements.

Kind regards,

Hudson Alves Pinto, Ph.D

Academic Editor

PLOS ONE

Additional Editor Comments (optional):

I congratulate the authors for their efforts in the revision. The new version improved a lot and the MS is now suitablefor publication.

---

## [Editor Report · Acceptance letter]

21 Mar 2023

PONE-D-22-21332R1 

Epidemiology of bovine schistosomiasis and associated risk factors in Ethiopia: A systematic review with meta-analysis of published articles, 2008-2018 

Dear Dr. Dagnaw:

I'm pleased to inform you that your manuscript has been deemed suitable for publication in PLOS ONE. Congratulations! Your manuscript is now with our production department. 

Kind regards, 

on behalf of

Dr. Hudson Alves Pinto 

Academic Editor

PLOS ONE